# HERV activation segregates ME/CFS from fibromyalgia while defining a novel nosologic entity

Karen Giménez-Orenga[1], Eva Martín-Martínez[2], Lubov Nathanson[3], Elisa Oltra[4]*

[1]Escuela de Doctorado, Catholic University of Valencia, Valencia, Spain; [2]National Health Service, Manises Hospital, Valencia, Spain; [3]Institute for Neuro-Immune Medicine, Dr. Kiran C. Patel College of Osteopathic Medicine, Nova Southeastern University, Fort Lauderdale, United States; [4]Department of Pathology, School of Health Sciences, Catholic University of Valencia, Valencia, Spain

## eLife Assessment

This **important** study substantially expands observations of HERV expression in the clinical settings. The evidence provided by the authors that HERV activity is an underlying etiological factor in ME/CFS and fibromyalgia is **compelling** and suggests further investigation into mechanisms. This work will be of broad interest to clinicians and researchers alike.

*For correspondence:
elisa.oltra@ucv.es

Competing interest: The authors declare that no competing interests exist.

**Abstract** Research of myalgic encephalomyelitis/chronic fatigue syndrome (ME/CFS) and fibromyalgia (FM), two acquired chronic illnesses affecting mainly females, has failed to ascertain their frequent co-appearance and etiology. Despite prior detection of human endogenous retrovirus (HERV) activation in these diseases, the potential biomarker value of HERV expression profiles for their diagnosis, and the relationship of HERV expression profiles with patient immune systems and symptoms had remained unexplored. By using HERV-V3 high-density microarrays (including over 350k HERV elements and more than 1500 immune-related genes) to interrogate the transcriptomes of peripheral blood mononuclear cells from female patients diagnosed with ME/CFS, FM, or both, and matched healthy controls ($n = 43$), this study fills this gap of knowledge. Hierarchical clustering of HERV expression profiles strikingly allowed perfect participant assignment into four distinct groups: ME/CFS, FM, co-diagnosed, or healthy, pointing at a potent biomarker value of HERV expression profiles to differentiate between these hard-to-diagnose chronic syndromes. Differentially expressed HERV–immune–gene modules revealed unique profiles for each of the four study groups and highlighting decreased γδ T cells, and increased plasma and resting CD4 memory T cells, correlating with patient symptom severity in ME/CFS. Moreover, activation of HERV sequences coincided with enrichment of binding sequences targeted by transcription factors which recruit SETDB1 and TRIM28, two known epigenetic silencers of HERV, in ME/CFS, offering a mechanistic explanation for the findings. Unexpectedly, HERV expression profiles appeared minimally affected in co-diagnosed patients denoting a new nosological entity with low epigenetic impact, a seemingly relevant aspect for the diagnosis and treatment of this prevalent group of patients.

## Introduction

Myalgic encephalomyelitis/chronic fatigue syndrome (ME/CFS), classified by the WHO with the ICD-11 8E49 code as a postviral fatigue syndrome, and fibromyalgia (FM) (ICD-11 MG30.0 for chronic primary pain) (*Harrison et al., 2021*), are chronic, disabling, acquired diseases, characterized by complex

symptomatology that affects multiple organs (*Bateman et al., 2021*; *Wolfe et al., 2010*). Diagnosis of ME/CFS and FM continues to be based on the clinical assessment of unspecific symptoms, such as debilitating fatigue, generalized pain, cognitive impairment or intestinal, sleep, and immune disturbances (*Carruthers et al., 2011*; *Carruthers et al., 2003*; *Wolfe et al., 2016*; *Wolfe et al., 2010*; *Wolfe et al., 1990*). Their frequent co-diagnosis drove the hypothesis of a single syndrome and promoted the search for common or differentiating factors (*Abbi and Natelson, 2013*; *Natelson, 2019*; *Wessely et al., 1999*). However, despite molecular support for ME/CFS and FM constituting different entities (*Groven et al., 2021*; *Light et al., 2012*; *Nepotchatykh et al., 2023*), their frequent joint appearance (over 50% in females) (*Castro-Marrero et al., 2017*) remains enigmatic.

Extensive research into ME/CFS and FM has not yet been able to ascertain their origin and pathophysiology. However, several environmental factors have been suggested as triggering agents (*Chu et al., 2019*; *Furness et al., 2018*; *Tschopp et al., 2023*). Viral infections are, particularly, gaining momentum after the emergence of a type of persistent post-viral syndrome with symptoms that closely resemble those in ME/CFS (*Komaroff and Lipkin, 2023*) and FM (*Clauw and Calabrese, 2024*), which has been defined as post-COVID-19 condition (ICD-11 RA02) (*Soriano et al., 2022*). The prevalence of post-COVID-19 cases meeting ME/CFS criteria has been estimated at 58% (*Jason and Dorri, 2022*) raising concerns of a health, social, and economic burden of unprecedented dimensions.

Viral infections trigger dramatic changes in host cells gene expression, affecting their metabolism and epigenetic landscape (*Liu et al., 2020*). Although functionally not well understood, these epigenetic changes involve de-repression of human endogenous retroviruses (HERVs) (*Macchietto et al., 2020*) sequences that were incorporated from exogenous viral infections during evolution and that currently represent about 8% of our genome (*Giménez-Orenga and Oltra, 2021*). In fact, SARS-CoV-2 infection alters peripheral blood mononuclear cell (PBMC) ERV expression in human, monkey, and mice expressing human ACE2 receptors, associating with immune-response activation and histone modification genes in severe COVID-19 cases (*Guo et al., 2024*), and the presence of HERV proteins in postmortem tissues of lungs, heart, gastrointestinal tract, brain olfactory bulb, and nasal mucosa from COVID-19 patients (*Charvet et al., 2023*).

Although still poorly understood, advances in sequencing technology have aided in the discovery of the wide range of physiological processes in which HERV participate, including coordination of the immune response (*Kassiotis, 2023*), interaction with microbiota (*Dopkins et al., 2022*; *Lima-Junior et al., 2021*), or shaping neurological functions (*Ferrari et al., 2021*), among other. Derangement of HERV expression associates with disease (e.g., multiple sclerosis; *Gruchot et al., 2023*), systemic lupus erythematosus (*Khadjinova et al., 2022*), or post-COVID-19 condition (*Giménez-Orenga et al., 2022*). Particularly, the expression of HERV-encoded proteins has been shown to stimulate an immune response present in autoimmunity (*Gruchot et al., 2023*; *Khadjinova et al., 2022*). In addition, HERV influence on pathophysiology extends to their long terminal repeats (LTRs) regulatory regions, shaping host gene expression (*Ito et al., 2017*).

Previous studies detected overexpression of some HERV families in immune cells of ME/CFS (*Rodrigues et al., 2019*) and FM (*Ovejero et al., 2020*) at the transcript level. However, the use of directional RT-qPCR (reverse transcription followed by quantitative polymerase chain reaction)-based approaches using degenerated primers limited the detection of affected genomic loci. In this study, we elevated these analyses to obtain genome-wide HERV profiles (HERV expression profiles) of patients having received ME/CFS, FM diagnosis, or both, hypothesizing that HERV expression profiles may reveal distinct underlying pathomechanisms that justify their classification as separate diseases, and/or common fingerprints explaining their joint high prevalence. To test our hypothesis, we scrutinized HERV and gene transcriptomes by using high-density microarray technology in a selected cohort of female patients carefully phenotyped by a single expert clinician into three defined groups of patients: patients fulfilling Canadian and International ME/CFS diagnosis criteria (*Carruthers et al., 2011*; *Carruthers et al., 2003*), patients fulfilling FM diagnosis ACR (American College of Rheumatology) criteria (*Wolfe et al., 2016*; *Wolfe et al., 2010*; *Wolfe et al., 1990*), or patients complying with both, ME/CFS and FM diagnosis criteria, being called co-diagnosed from now on. The results of comparing HERV profiles within PBMCs across these three patient groups as well as to HERV profiles of matched healthy subjects allowed, not only for the separation of patients from healthy individuals but also for perfect discrimination of patients into three distinct disease groups, suggesting distinct subjacent pathomechanisms. Unexpectedly, HERV expression profiles appeared minimally affected

in co-diagnosed patients denoting a new nosological entity with low epigenetic impact. In addition, HERV profiling also exposed some commonalities between the FM and ME/CFS subgroups and marked quantitative differences within the ME/CFS group that correlated with immune disturbances and patient symptomatology, aspects that may well set new criteria for the differential diagnosis of FM and ME/CFS as well as for patient subtyping with expected impact in precision medicine programs.

## Results

### Demographics and clinical characteristics of participants

The study compared a total of 43 female subjects: 8 ME/CFS cases, 10 FM, 16 co-diagnosed cases, and 9 matched healthy controls (*Figure 1*). The average age for participants was 54 ± 3 years (range 50–58) for ME/CFS subjects, 50 ± 5 years (range 42–58) for FM, 47 ± 15 years (range 22–70) for co-diagnosed individuals, and 51 ± 6 (range 43–61) for controls (*Figure 1—figure supplement 1A*, *Supplementary file 1A*). Questionnaires used for patient symptom phenotyping included Fibromyalgia Impact Questionnaire (FIQ) (*Burckhardt et al., 1991*), Multi Fatigue Inventory (MFI) (*Smets et al., 1995*) for general fatigue, and Short-Form-36 Health Survey (SF-36) (*Mchorney et al., 1993*) for quality-of-life assessment (*Figure 1—figure supplement 1A*). Average total and subdomain scores are shown in *Table 1* for each patient group, while itemized questionnaire scores are provided in *Supplementary file 1A*. According to total FIQ scores, after considering that a score <39 indicates mild affection, ≥39 through ≤59 is assigned to moderately affected patients, and >59 to severe affection (*Rivera and González, 2004*) around 88% of cases in the studied cohort correspond with severe cases while 12% are moderate. No statistically significant difference between patient groups was observed for any questionnaire domain except for MFI Physical Fatigue (p = 0.011) and Reduced Motivation (p = 0.038), and only between the ME/CFS and co-diagnosed groups (*Supplementary file 1B*).

### HERV signatures discriminate ME/CFS, FM, and co-diagnosed groups from healthy controls

Genome-wide HERV expression profiles for each of the four study groups (three disease groups: ME/CFS, FM, and co-diagnosed, plus one control group corresponding to healthy participants), using custom high-density Affymetrix HERV-V3 microarrays (*Becker et al., 2017*), showed that a set of 489 HERV (502 probesets) is differentially expressed (DE) between at least two of the groups (FDR <0.1 and |log₂FC| >1) (*Figure 1*; *Supplementary file 1C*), confirming dysregulation of particular sets of HERV elements in the immune systems of ME/CFS, FM, and co-diagnosed cases as compared to healthy controls. Volcano plots show the number of HERV elements found significantly over- or underexpressed for each of the comparisons, evidencing an enhanced dysregulation of HERV in ME/CFS as compared to FM, co-diagnosed, or healthy control groups (*Figure 1A*). Consistently, hierarchical clustering of samples based on DE HERV fingerprints clearly differentiate these pathologies from healthy subjects, strikingly segregating ME/CFS from all other compared groups (*Figure 1B*); while hierarchical clustering of DE HERV sequences revealed four main clusters consisting of loci specifically overexpressed (Cluster 1, 251 HERV, 293 probesets) or underexpressed (Cluster 2, 119 HERV, 163 probesets) in ME/CFS, loci specifically overexpressed in FM (Cluster 3, 68 HERV, 74 probesets), and loci underexpressed in all three pathologies (Cluster 4, 54 HERV, 64 probesets). DE of the HERV loci for each study group was corroborated by analysis of HERV expression levels encompassed in each cluster, as illustrated with violin plots (*Figure 1B*), with some of the changes being validated by the alternative RT-qPCR approach, as shown in *Figure 1—figure supplement 1B*.

Closer examination of Clusters 1 and 2 indicated that the ME/CFS group could be subdivided into two subgroups showing differential levels of HERV expression profiles (*Figure 1C*). While both ME/CFS subgroups (subgroups 1 and 2) show clear upregulation (Cluster 1, p < 0.0001) or downregulation (Cluster 2, p < 0.0001) of several HERV loci as compared to all the other study groups, they show quantitative differences, with the ME/CFS subgroup 2 exhibiting a more pronounced dysregulation than subgroup 1, particularly affecting the downregulated Cluster 2 (*Figure 1C*). Interestingly, subgroup 2 ME/CFS patients present more accused levels of fatigue (p < 0.05) and reduced motivation as compared to the co-diagnosed group and the ME/CFS subgroup 1 (*Supplementary file 1D*)

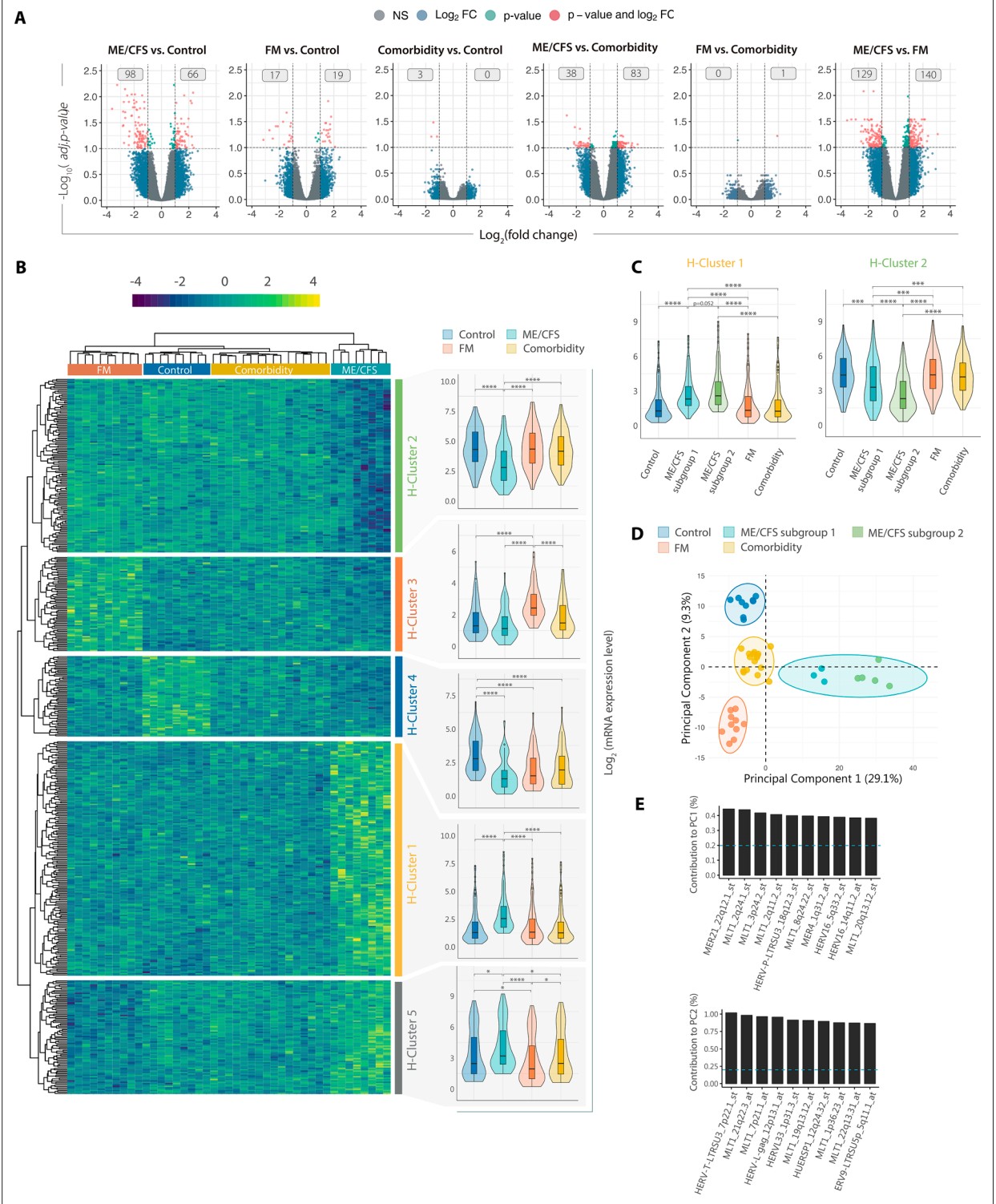

**Figure 1.** DE of HERV elements discriminates myalgic encephalomyelitis/chronic fatigue syndrome (ME/CFS), fibromyalgia (FM), and co-diagnosis. (**A**) Volcano plots showing log₂(fold changes) and the adjusted p-values for all HERV assessed with HERV-V3 microarray for each set of groups, as indicated. Red dots indicate DE HERV (FDR <0.1 and |log₂FC| >1). Gray boxes show numbers of overexpressed or underexpressed HERV elements. (**B**) HERV expression heatmap and cluster analysis of ME/CFS (*n* = 8, green), FM (*n* = 10, orange), co-diagnosed (*n* = 16, yellow), and healthy control (*n* = 9, blue) samples. The heatmap includes all HERV probes displaying significant DE between at least two of the compared groups (FDR <0.1 and |log₂FC| >1). Clusters 1–4 correspond to groups of HERV probes displaying significant over/under-expression in at least one group. Box and violin plots summarize the distribution and expression levels of the DE probes in each cluster per study group. The scaled mean expression value (*z*-score) for each HERV

*Figure 1 continued on next page*

*Figure 1 continued*

probe is plotted. Box plots show the median *z*-score value and the first and third quartiles. (**C**) Box and violin plots for Clusters 1 and 2 differentiating ME/CFS subgroups 1 and 2. (**D**) Principal component analysis of DE HERV. (**E**) Bar plot of top 10 DE HERV with greater influence/contribution to principal component 1 (PC1) or principal component 2 (PC2). Statistical tests: unpaired two-sample Wilcoxon test with Benjamini–Hochberg p-value correction (***p < 0.001, ****p < 0.0001).

The online version of this article includes the following figure supplement(s) for figure 1:

**Figure supplement 1.** Differences between patients and controls.

indicating that quantitative HERV fingerprint differences might help to assess the degree of ME/CFS severity.

In line with our findings, unsupervised principal component analysis (PCA) of DE HERV loci supports perfect discrimination of samples by study group and differentiates the two identified ME/CFS subgroups (*Figure 1D*). While principal component 1 (PC1) perfectly segregates the two ME/CFS subgroups from FM, co-diagnosed, and control groups; principal component 2 (PC2) allows partition of FM, co-diagnosed, and control samples, but not so much of ME/CFS from co-diagnosed cases. Thus, PC1 explains better the variance observed across groups (29.1%) than PC2 (9.3%), suggesting a prominent role of DE HERV loci in ME/CFS pathology. Top 10 contributing HERVs to principal components

**Table 1.** Patient health status assessment with FIQ, MFI, and SF-36 (*Burckhardt et al., 1991*; *Mchorney et al., 1993*; *Smets et al., 1995*) questionnaires.

| | Control (*n* = 9)<br>Mean ± SD [range] | ME/CFS (*n* = 8)<br>Mean ± SD [range] | FM (*n* = 10)<br>Mean ± SD [range] | Co-diagnosed (*n* = 16)<br>Mean ± SD [range] |
|---|---|---|---|---|
| Age | 51 ± 2 [43–61] | 54 ± 3 [50–58] | 50 ± 5 [42–58] | 47 ± 15.48 [22 – 58] |
| BMI | 25.03 ± 2.07 [22.10–28.04] | 26.29 ± 3.33 [21.34–31.22] | 25.31 ± 1.73 [23.68–28.40] | 24.80 ± 4.72 [18.71–30.43] |
| FIQ | | | | |
| Total FIQ | 28.16 ± 13.08 [0–2.86] | 74.4 ± 4.8 [56.3–85.6] | 75.9 ± 3.4 [51.6–92.8] | 74.6 ± 2.5 [47.8–96.3] |
| Function | 2.38 ± 0.57 [1.98–3.63] | 5.2 ± 2.2 [1.7–7.9] | 6.8 ± 1.5 [4.3–8.9] | 5.2 ± 1.8 [1.7–9.2] |
| Overall | 10.01 ± 0 [10.01–10.01] | 8.9 ± 3.0 [1.4–10.0] | 8.2 ± 3.4 [0–10.0] | 7.9 ± 2.7 [1.4–10] |
| Symptoms | 0.32 ± 0.95 [0–2.86] | 6.6 ± 4.8 [0–10.0] | 6.0 ± 3.4 [0–10.0] | 6.2 ± 2.5 [2.9–10] |
| MFI | | | | |
| General fatigue | 11.56 ± 4.19 [5–17] | 17.6 ± 3.1 [11–20] | 16.3 ± 4.6 [10–20] | 13.6 ± 3.2 [7–20] |
| Physical fatigue | 10.11 ± 4.01 [6–17] | 18.0 ± 1.9 [16–20] | 17.0 ± 3.7 [12–20] | 14.1 ± 2.9 [12–20] |
| Reduced activity | 7.33 ± 2.74 [4–12] | 15.9 ± 3.4 [12–20] | 15.9 ± 4.4 [9–20] | 12.9 ± 2.8 [11–20] |
| Reduced motivation | 7.22 ± 3.31 [4–15] | 15.4 ± 3.7 [9–19] | 14.8 ± 4.1 [9–20] | 11.6 ± 2.4 [8–16] |
| Mental fatigue | 7.11 ± 2.93 [4–13] | 15.5 ± 3.7 [11–20] | 15.1 ± 3.8 | 13.3 ± 3.1 [9–19] |
| SF-36 | | | | |
| Physical functioning | 86.67 ± 13.23 [65–100] | 44.5 ± 14.5 [15–60] | 31.5 ± 13.9 | 35.3 ± 14.8 [0–65] |
| Role physical | 83.33 ± 17.95 [50–100] | 3.2 ± 8.8 [0–25] | 10.6 ± 17.7 [0–43.8] | 14.5 ± 21.9 [0–75] |
| Bodily pain | 58.61 ± 12.63 [45–80] | 19.3 ± 11.7 [0–32.5] | 19.3 ± 17.8 [0–45] | 17.3 ± 14.2 [0–45] |
| General health | 65.56 ± 15.09 [35–85] | 18.9 ± 11.0 [3.8–35.0] | 24.8 ± 16.1 [0–50] | 23.4 ± 14.8 [0–45] |
| Vitality | 56.94 ± 9.08 [43.75–75.00] | 13.6 ± 10.1 [0–30] | 11.1 ± 16.8 [0–43.8] | 14.5 ± 11.6 [0–40] |
| Social functioning | 84.72 ± 13.66 [62.50–100] | 33.6 ± 21.9 [4.0–77.5] | 25.0 ± 23.2 [0–62.1] | 29.5 ± 25.6 [0–75] |
| Role emotional | 79.63 ± 20.88 [50–100] | 12.8 ± 35.3 [0–100] | 43.3 ± 37.2 [0–100] | 12.8 ± 38.9 [0–100] |
| Mental health | 71.67 ± 18.54 [40–95] | 34.9 ± 21.2 [3.4–72.0] | 44.7 ± 19.9 [12–70] | 45.2 ± 17.1 [15–80] |

FIQ, Fibromyalgia Impact Questionnaire; MFI, Multi Fatigue Inventory; SF-36, Short-Form 36 Health Survey; SD, standard deviation; SE, standard error. Range refers to the possible values in the studied group.

PC1 and PC2 are shown (*Figure 1E*). In summary, these results show that ME/CFS, FM, and their co-diagnosis present unique HERV expression profiles capable of discriminating patient subtypes from healthy subjects, providing potential biomarkers for their differential diagnosis and stratification. ME/CFS shows the greatest HERV dysregulation which, importantly, associates with patient symptomatology, suggesting a prominent role of DE HERV in the pathomechanism of this disease, while the co-diagnosed group presents the closest HERV expression profiles to those of healthy controls indicating much less involvement of epigenetic changes in these patients, thus, defining a different nosological entity from ME/CFS or FM.

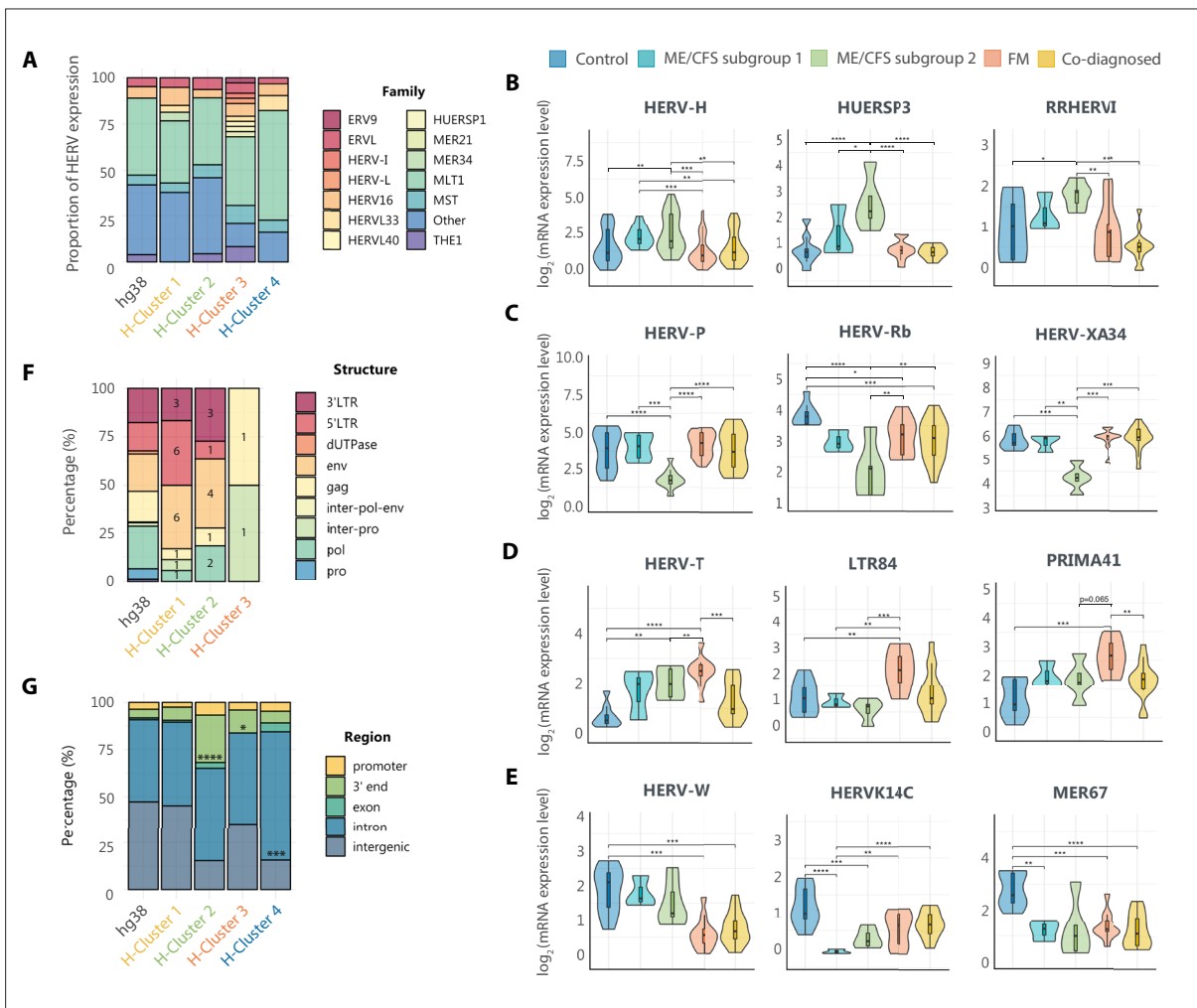

**Figure 2.** Diverse solitary long terminal repeat (LTR) families are deregulated in myalgic encephalomyelitis/chronic fatigue syndrome (ME/CFS), fibromyalgia (FM), and co-diagnosed conditions. (**A**) Relative contribution of HERV families to each cluster, calculated as the proportion of HERV loci assigned to each family relative to total HERV loci in each cluster. Families with a representation of at least 2.5% are shown. Main HERV families (**B**) downregulated in ME/CFS, FM and co-diagnosed groups (Cluster 4), (**C**) upregulated in ME/CFS (Cluster 1), (**D**) downregulated in ME/CFS (Cluster 2), and (**E**) upregulated in FM (Cluster 3). Box and violin plots summarize distributions and expression levels of the different HERV probes belonging to the same family in the different study groups. The scaled mean expression value (z-score) for each HERV probe is plotted. Box plots show the median z-score value and the first and third quartiles. (**F**) Proportion of HERV subdomains expressed by cluster. (**G**) Genomic context of the DE HERV loci by cluster. Statistical tests: Fisher's exact test, *t*-test or Wilcoxon test with Benjamini–Hochberg p-value correction (*p < 0.05, **p < 0.01, ***p < 0.001, ****p < 0.0001).

The online version of this article includes the following figure supplement(s) for figure 2:

**Figure supplement 1.** DE HERV families by study group.

**Figure supplement 2.** DE HERV structures by study group.

**Figure supplement 3.** DE HERV genomic location by study group.

**Figure supplement 4.** Intergenic solo long terminal repeat (LTR) DE across study groups.

## Dysregulated HERV families are disease specific

Since preferential HERV-family derangements may indicate particularly altered functions in disease (*Ito et al., 2017*), we next examined the family composition of the identified DE HERV clusters (*Figure 2*). Overall, ME/CFS (Clusters 1 and 2) outstands with the largest heterogeneity with up to 66 HERV dysregulated families, contrasting with only 22 families in FM (Cluster 3) and 16 in Cluster 4, the latter representing DE HERV common to all patient groups (*Supplementary file 1E*, *Figure 2—figure supplement 1*).

Among DE HERV families 8 appear to be common across all clusters, and 6 of them represent the most abundantly deregulated (>2.5%) (MLT1, MST, THE1, HERV16, ERVL, and HERVL33) (*Figure 2A*, *Supplementary file 1E*), suggesting an unspecific association with all disease groups. Within these families, however, MLT1 appears as the only significantly downregulated in all patient groups (p < 0.01) (*Figure 2—figure supplement 1*), accounting for more than a third of the DE HERV in each cluster (*Figure 2A*, *Supplementary file 1E*), a finding that may be, at least partially, due to the fact of MLT1 being the most abundant HERV family in the human genome (*Figure 2A*).

Interestingly, unique sets of DE HERV families were found for each disease (*Supplementary file 1E*). Up to 21 families, comprising HERV-H, HUERSP3, and RRHERVI among other, appeared specifically upregulated in Cluster 1 (*Figure 2B*, *Figure 2—figure supplement 1*, and *Supplementary file 1E*), while 12 families, including HERV-P, HERV-Rb, and HERV-XA34, were specifically silenced in Cluster 2, both strikingly DE in ME/CFS patients from subgroup 2 (*Figure 2C*, *Figure 2—figure supplement 1*, *Supplementary file 1E*). Regarding FM, three families consisting of HERV-T, LTR84, and PRIMA41, were found specifically upregulated (Cluster 3) (*Figure 2D*, *Figure 2—figure supplement 1*, *Supplementary file 1E*). Five families were commonly downregulated across ME/CFS, FM, and co-diagnosed groups, including HERV-W, HERVK14C, and MER67 (*Figure 2E*, *Figure 2—figure supplement 1*, *Supplementary file 1E*). Contrary to the literature (*Ovejero et al., 2020*), reporting upregulation of HERV-W family in FM, our results showed downregulation of HERV-W family in all patient groups, more accentuated in FM and co-diagnosed groups than ME/CFS with respect to healthy levels (*Figure 2E*). In all, these results reveal family-specific derangement of HERV expression in immune cells of FM and ME/CFS but not in co-diagnosed cases.

In light of the potential involvement of HERV-encoded proviral proteins in disease development (*Giménez-Orenga and Oltra, 2021*) and to deepen the analysis of disease-specific HERV fingerprints, we then determined the proportion of these HERV fingerprints corresponding to solitary and proviral elements, as well as the different subdomains of the HERV structure according to probe detection in the microarrays that is, LTR containing regulatory sequences versus gag, pol, and env protein-coding genes. Although functional information of these sequences is quite limited, the nature of the detected HERV motifs may inform of potential regulatory mechanisms leading to or perpetuating disease status (e.g., toxic env proteins or non-coding regulatory sequences). In agreement with hg38 HERV genomic data, it was found that the great majority of the dysregulated HERV loci corresponded to solitary LTR elements, which are known to be the most abundant in our genome, with few exceptions corresponding to proviral HERV (*Figure 2F*, *Figure 2—figure supplement 2*).

Lastly, given the potential of HERV LTRs to influence the expression of neighboring genes through the regulatory elements they hold, we examined the genomic context of the DE HERV loci by cluster (*Figure 2G*, *Figure 2—figure supplement 3*). Although not allocating significant association to promoter regions, repressed loci in ME/CFS (Cluster 2) and overexpressed loci in FM (Cluster 3) were significantly associated with gene 3' endings, and silenced loci in the three pathologies (Cluster 4) significantly associated with intronic regions. These results suggest that the molecular mechanisms by which disease-specific deregulated HERV loci could be participating in the development or maintenance of disease likely involve changes in transcript turnover or availability of alternative splicing events. In addition, DE analysis of intergenic solo LTRs show that they are underexpressed in ME/CFS (*Figure 2—figure supplement 4*), pointing at the participation of epigenetic regulatory mechanisms in the disease.

## HERV fingerprints associate with abnormal immune gene expression

Since disease-associated HERV derangement could indicate, even mediate, differences in global gene expression, we proceeded to analyze potential associations between disease-associated HERV fingerprints and a set of genes involved in eight pathways (immunity, inflammation, cancer, central nervous

system disorders, differentiation, telomere maintenance, chromatin structure, and gag-like genes) detected by probes present in this same HERV-V3 high-density microarrays (*Becker et al., 2017*). Differential transcript analysis revealed 368 genes (1037 probesets) DE between at least two study groups (FDR <0.1 and |log₂FC| >1) (*Supplementary file 1F*). In line with DE HERV expression, ME/CFS immune cells showed a more accused dysregulation in gene expression profiles compared to FM's and co-diagnosed groups (*Figure 3*, *Figure 3—figure supplement 1A*). Hierarchical clustering of samples and PCA based on DE genes, however, did not cluster all samples by study group as HERV signatures did, with the only exception of ME/CFS subgroup 2, which exhibited a markedly different gene expression profile further supporting its distinction from the remaining study groups (*Figure 3—figure supplement 1B*).

To uncover potentially affected physiologic functions linked to DE HERV, we examined how DE HERVs and DE genes with similar expression patterns grouped together in modules based on their intrinsic relationships by their hierarchical co-clustering (*Figure 3*). Then, the functional significance of these modules was assessed by gene ontology (GO) analysis of the DE genes within each module. The hierarchical clustering analysis resulted in the identification of eight distinct modules, each characterized by unique combinations of DE HERV and DE gene patterns across all four study groups (*Figure 3*). In particular, 37 HERV (19 from Module 1 and 18 from Module 6) with decreased expression levels across the three disease groups, correlated with several genes involved in the differentiation of immune cells and regulation of cell–cell adhesion (*Figure 3A*). Remarkably, ME/CFS exhibited upregulation of 258 HERV loci with expression levels that correlated with 34 genes involved in alpha-beta T cell activation and T-helper 17 cell commitment-related genes (Module 3). In addition, 78 HERV downregulated in ME/CFS correlated with pathogen detection-related genes (*Figure 3B*, GO:0016045, GO:0016032, GO:0098543), suggesting a special involvement of the immune system in the disease. Lastly, several HERV and genes exhibited upregulated expression in FM while appearing downregulated in ME/CFS. These FM-specific DE expressed genes are involved in telomere structure organization and maintenance (Modules 2 and 5), cellular response to glucocorticoids (Module 7), and regulation of NF-κB signaling (Module 8) (*Figure 3C*), indicating divergent immune states in ME/CFS and FM with a differential impact in inflammatory processes.

## Co-expression of DE HERV and immune-response genes predominantly occur by mechanisms other than co-transcription

To further investigate the relationship between HERV dysregulation and immune abnormalities, we mapped the genomic localization of HERV and genes within each module, looking for genomic overlaps that could explain their correlation by co-expression. Surprisingly, the results showed no major overlaps of DE HERV and their correlated DE genes (*Figure 4A*), with distances that separate them at least 100 kbp (*Figure 4B*, *Supplementary file 1G*), thereby suggesting overall independent expression of DE HERV and DE genes. As an exception, it was noticed that some elements from the MLT1 family co-localized with DE genes within a 30 kbp window (*Supplementary file 1G*). For example, the element MLT1_5q32, from Module 1, appears located 18,114 bp upstream of the *CD74* gene, involved in MHC class II antigen processing, or the element MLT1_4q24, belonging to Module 6, lies within *NFκB1* first intron. Both HERV loci and their co-localized genes exhibited downregulation in ME/CFS, FM, and co-diagnosed groups (p < 0.05). Likewise, an exception was noticed for the MLT1_8q12.1 and MLT1_Xp22.3 HERV elements in Module 2, located within *LYN*'s tenth intron and *TLR8*-long isoform's second exon, respectively. Their genomic co-localization may explain that their specific lower expression levels in the ME/CFS subgroup 2 occur as result of co-transcription (*Figure 4C*).

Since common *trans*-regulatory mechanisms may explain correlation by co-expression in the absence of co-transcription, we analyzed DE HERV in each module for potential enrichment in transcription factor binding sites (TFBS) by comparing the annotated TFBS within dysregulated HERV genomic regions against the full set of TFBS in the genome (based on publicly available human ChIP-seq data provided by ReMap2022) (*Figure 4D*). Then, we performed GO analysis for HERV-gene sets enriched TFBS to infer their function (*Figure 4—figure supplement 1*). Interestingly enough, the results showed enrichment of binding sites for transcription factors involved in the immune response in Modules 1 through 7, further supporting the role of immune disturbances in ME/CFS, FM, and co-diagnosed cases, while no significant enrichment of TFBS was found in Module 8. In addition, HERV

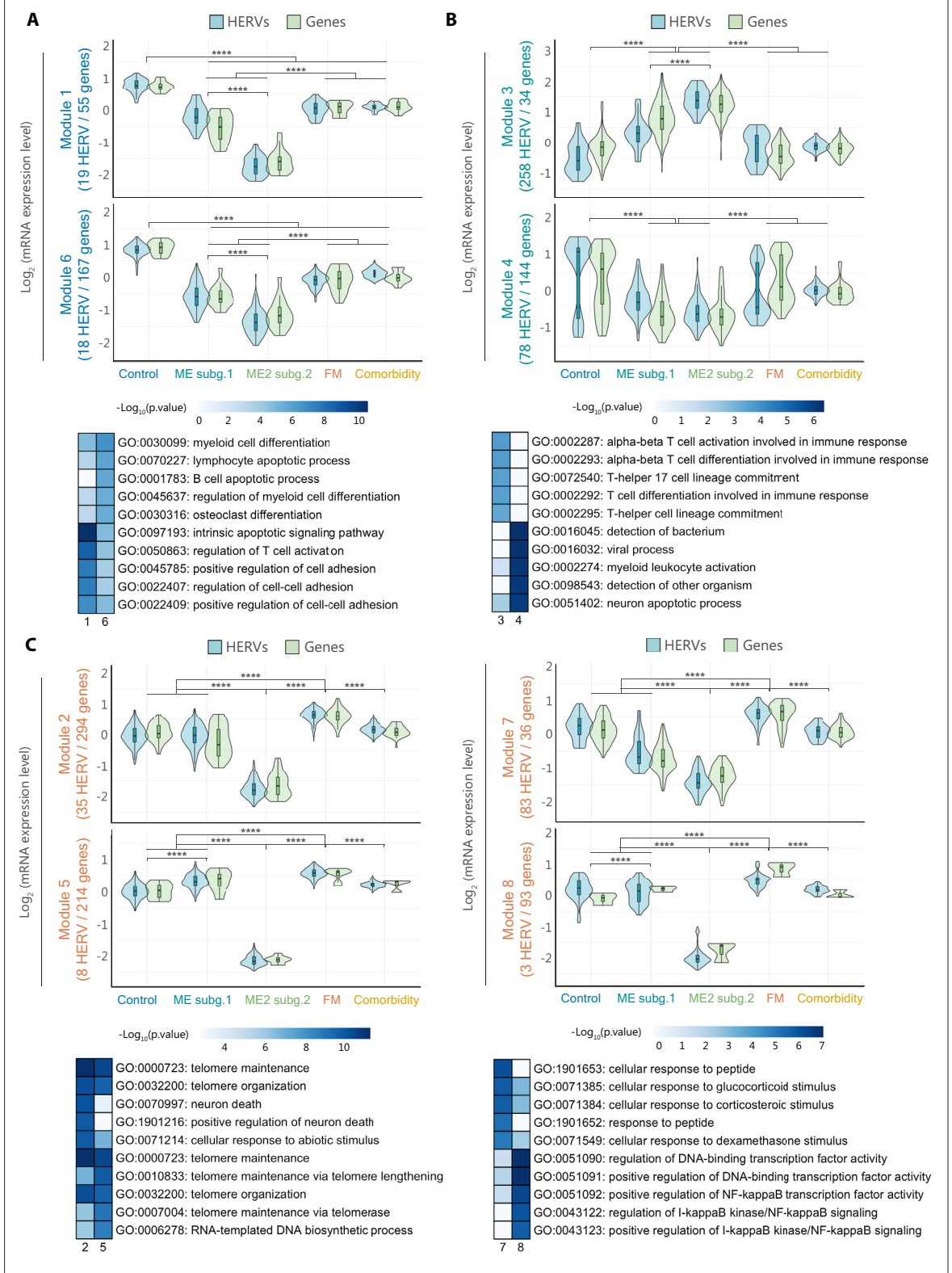

**Figure 3.** HERV expression correlates with immune-response genes. Hierarchical clustering of DE HERV and genes (FDR <0.1 and |log₂FC| >1) providing eight modules with highly correlated expression levels. Box and violin plots summarize the distribution and expression level of the different HERV (blue) and gene (green) probes in Modules (**A**) 1 and 6, (**B**) 3 and 4, and (**C**) 2, 5, 7, and 8, per study group. The scaled mean expression value (z-score) for each HERV and gene probe sets are plotted. Box plots show the median z-score values with first and third quartiles. Statistical tests: Wilcoxon test (****p < 0.0001).

*Figure 3 continued*

The online version of this article includes the following figure supplement(s) for figure 3:

**Figure supplement 1.** Differential gene expression for myalgic encephalomyelitis/chronic fatigue syndrome (ME/CFS), fibromyalgia (FM), and comorbidity groups.

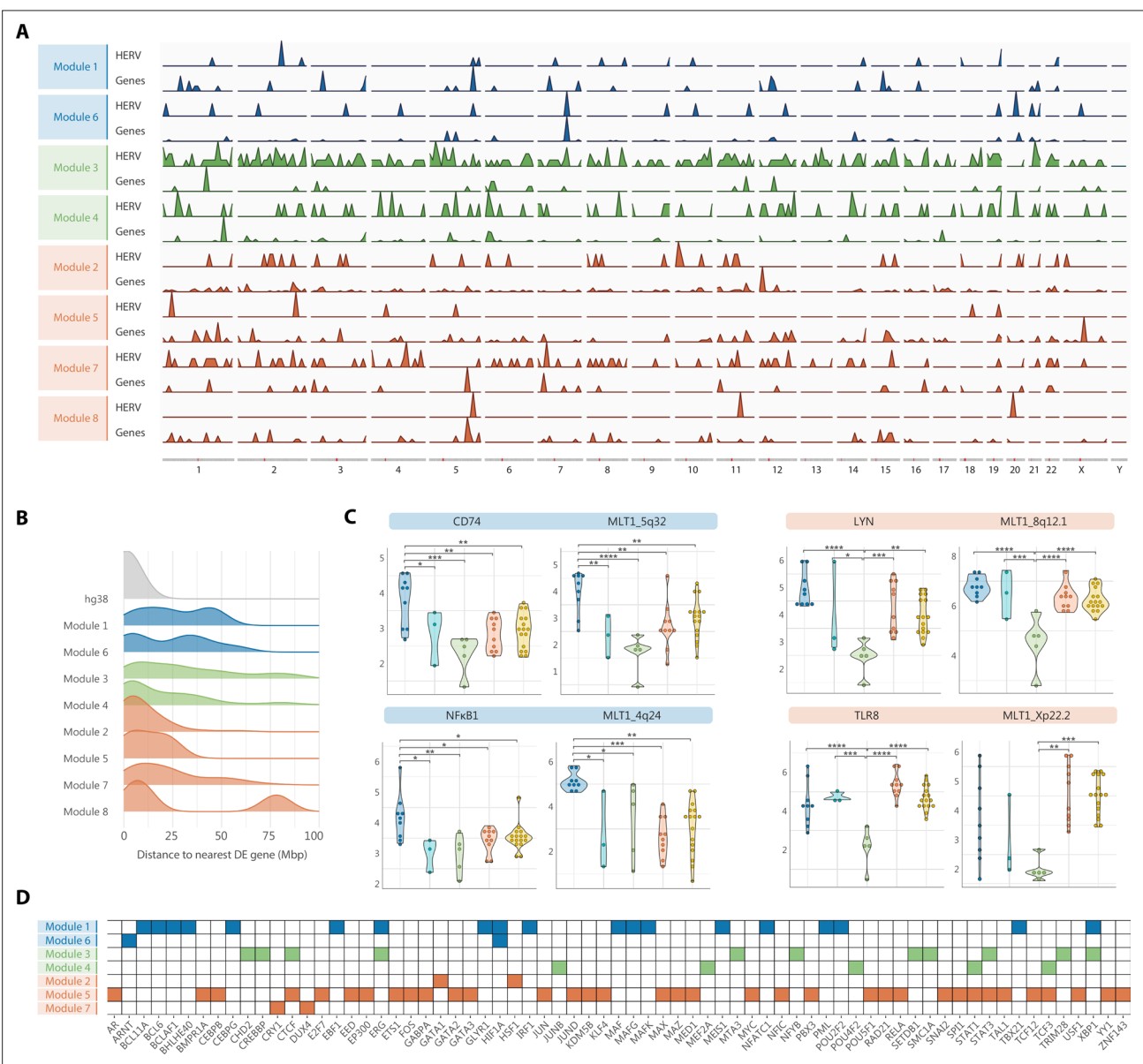

**Figure 4.** DE HERV sequences and DE immune-response genes are majorly independently transcribed. (**A**) Genomic distribution of HERV, and genes encompassed in Modules 1 through 8. (**B**) Genomic distance between DE HERV and their nearest DE genes per module, plotted as density curves illustrating the distribution of distances from 0 to 100 million base pair (Mbp). (**C**) Violin dot plots showing the distribution and expression levels of some examples of co-localized HERV loci and genes in myalgic encephalomyelitis/chronic fatigue syndrome (ME/CFS) subgroup 1 (*n* = 3), ME/CFS subgroup 2 (*n* = 5), FM (*n* = 10), co-diagnosed (*n* = 16), and healthy control (*n* = 9) samples. Blue and orange highlighted headings indicates the pair HERV/gene belonged to Modules 1 or 6, or Modules 2, 5, 7, or 8, respectively. (**D**) Heatmap of enriched transcription factor binding sites (TFBS) in HERV loci of Modules 1 through 7. Module 8 did not show significant enrichment of TFBS. Statistical tests: Wilcoxon test (*p < 0.05; **p < 0.01, ***p < 0.001, ****p < 0.0001).

The online version of this article includes the following figure supplement(s) for figure 4:

**Figure supplement 1.** Gene ontology analysis of transcription factors binding sites enriched in HERV loci of Modules 1–7.

downregulated across the three patient groups (Modules 1 and 6) were enriched in TFBS involved in angiogenesis, while those upregulated in FM but downregulated in ME/CFS (Modules 2, 5, and 7), were enriched in TFBS for miRNA transcription and glucocorticoid signaling. It is worth mentioning that HERV upregulated in ME/CFS (Module 3) were markedly enriched with binding sites for chromatin remodeling factors, from which SETDB1 and TRIM28 stand out as key epigenetic repressors of HERV expression and CTCF for its importance in the establishment of topologically associating domains in chromosomes. These results therefore support the possibility that DE HERV expression occurs independently, but in coordination with their correlated DE genes by yet to define mechanisms, likely driven by and/or driving epigenetic changes in immune cells of ME/CFS patients.

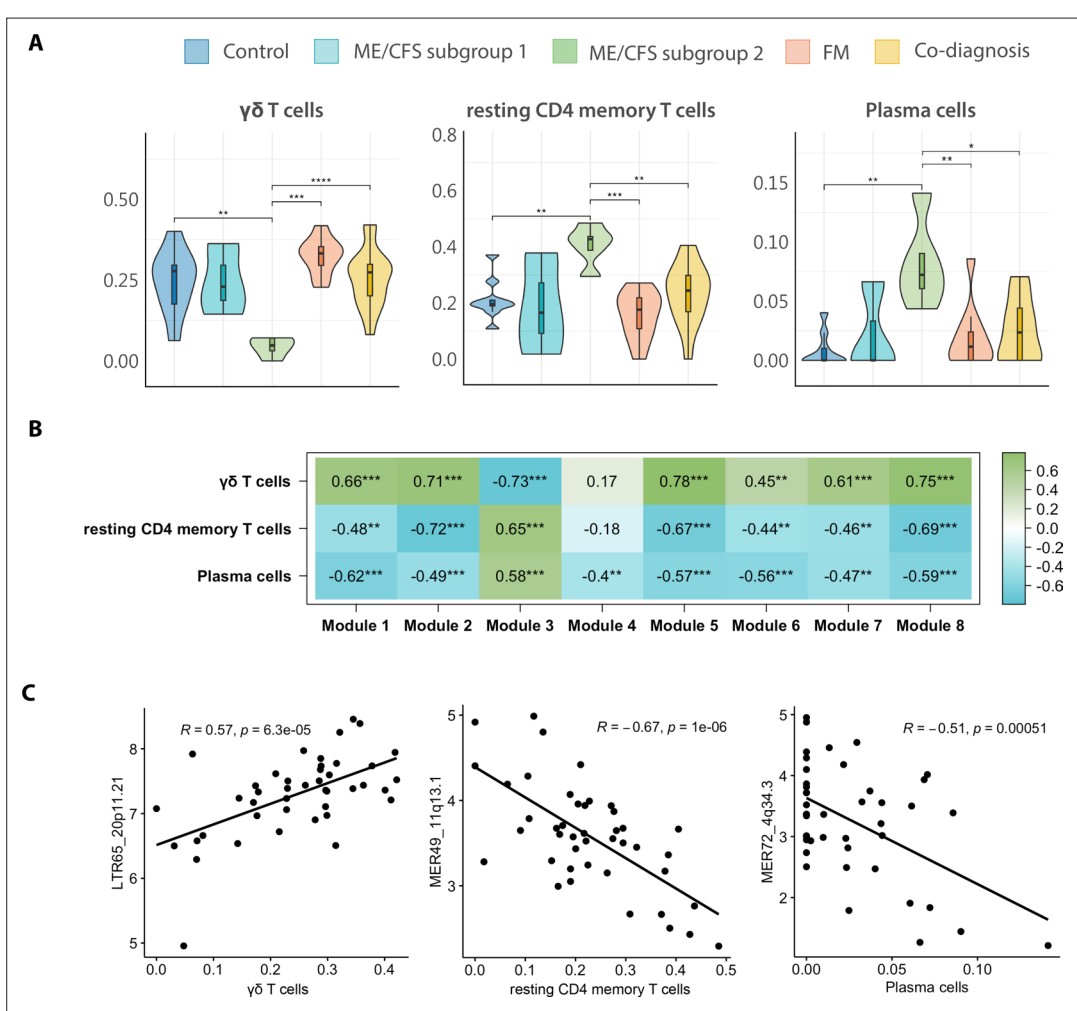

**Figure 5.** HERV fingerprints correlate with immune cell profiles. (**A**) Measurements of immune cell proportions by CIBERSORTx per study group ($n = 8$ myalgic encephalomyelitis/chronic fatigue syndrome [ME/CFS], $n = 10$ fibromyalgia [FM], $n = 16$ co-diagnosed, $n = 9$ controls). (**B**) Association between modules and immune cell proportions was evaluated by correlating eigengenes from each module with cell proportion values obtained from CIBERSORTx analysis. Boxes show Pearson correlation values and associated p-values (*p < 0.05, **p < 0.01, ***p < 0.001, ****p<0.0001) between gene expression levels of each module and quantity of cells from each specific type. A value of 1 (green) and –1 (blue) quantify strongest positive and negative correlations, respectively, while 0 (white) shows no correlation. (**C**) Scatter plots between top HERV loci expression levels and immune cell proportion as measured by CIBERSORTx. Pearson correlation coefficients and associated p-values are shown in plots. Correlation 95% confidence intervals are: 0.33–0.74 for LTR65_20p11.21 versus γδ T cells, –0.81 to –0.46 for MER49_11q13.1 versus resting CD4 memory T cells, and –0.70 to –0.25 for MER72_4q34.3 versus plasma cells.

## DE HERV strongly associates with plasma and T-cell levels

Given the detected relationships between HERV dysregulation and immunological disturbances, we investigated differences in immune cell populations across the study groups by performing CIBER-SORTx analysis on normalized microarray gene expression data. CIBERSORTx is an analytical tool that provides an estimation of cell type abundance using gene expression data (*Newman et al., 2019*). The results point at a decrease in γδ T cells as well as an increase in resting CD4 T memory and plasma cells (p < 0.05) in ME/CFS patients from subgroup 2 (*Figure 5A*). To further understand the relationship between gene signatures and the potential differences in immune cell populations with DE HERV, we analyzed correlations across HERV and genes included in Modules 1 through 8 and immune cell proportions, finding that dysregulated HERV loci moderately to strongly correlate with plasma cell and resting CD4 memory T cells increase, and γδ T cell population decrease (|R| > 0.4, p < 0.01) (*Figure 5B*). Curiously, HERV and genes upregulated in ME/CFS subgroup 2 (Module 3), negatively correlate with γδ T cell abundance (e.g., LTR65_20p11.21, R = 0.57, p = 6.3e−05) and positively correlate with resting CD4 memory T cells, and plasma cell abundances (e.g., LTR65_20p11.21, R = 0.57, p = 6.3e−05; MER49_11q13.1, R = −0.67, p = 1e−06) (*Figure 5C*), while the tendency of correlation appears reversed for the other modules. These results suggest a potential relationship between HERV expression and immune cell fractions in a subgroup of ME/CFS patients.

## HERV and immune alterations associate with ME/CFS diagnosis

Lastly, in an effort to detect potential relationships between deregulated HERV and genes, and patient diagnosis, we analyzed the association between DE of HERV and genes belonging to each module

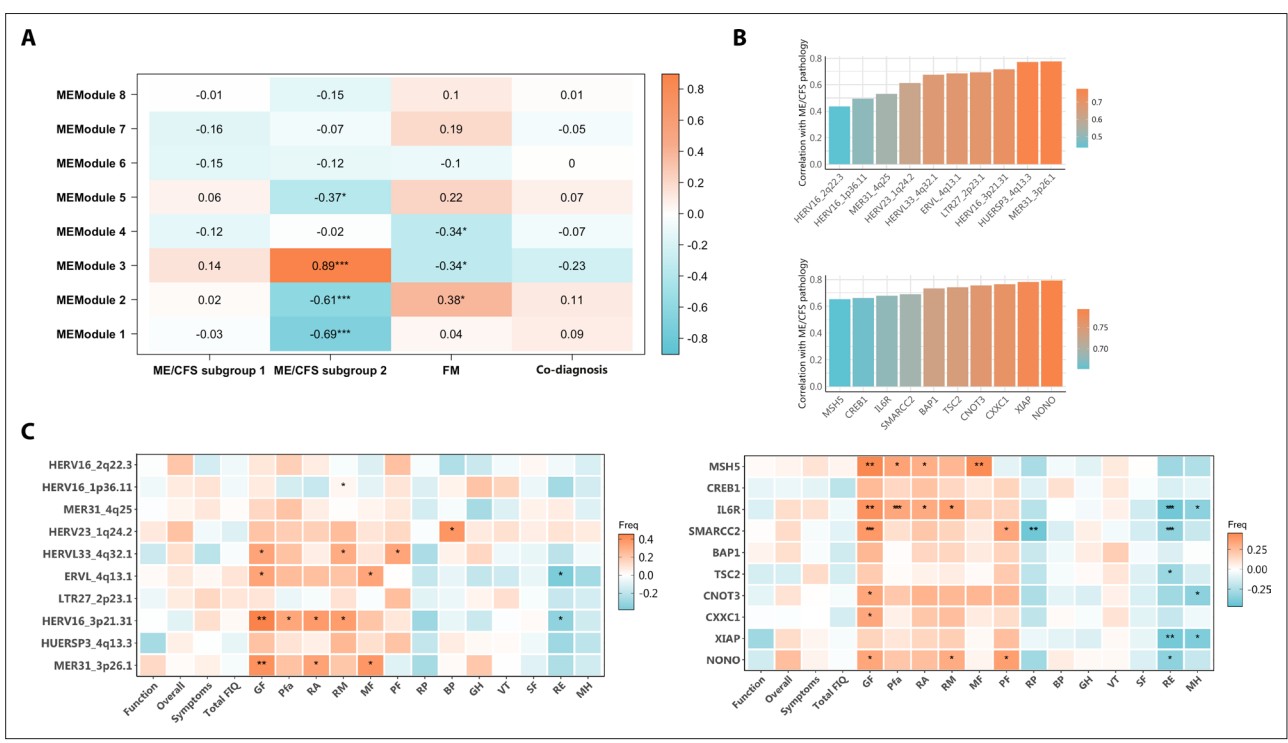

**Figure 6.** HERV and gene deregulation associates with disease symptomatology. (**A**) Association between modules and disease determined by correlations of eigengenes from each module with binarized disease traits. Boxes show Pearson correlation values and associated p-values (*p < 0.05, **p < 0.01, ***p < 0.001) by module and each disease trait. A value of 1 (orange) and –1 (blue) quantify strongest positive and negative correlations, respectively, while 0 (white) shows no correlation. (**B**) Barplots showing top 10 HERV loci (upper) and top 10 genes (lower) with highest correlations with myalgic encephalomyelitis/chronic fatigue syndrome (ME/CFS) disease. Bar height and color display the strength of each correlation, being high orange bars more correlated than lower blue bars. (**C**) Association between top 10 HERV loci (left) and top 10 genes (right) with disease traits as determined by their correlation with ME/CFS patient symptoms as assessed by questionnaire scores. Boxes show Pearson correlation values and associated p-values (*p < 0.05, **p < 0.01) by DE HERV or DE gene as indicated for Fibromyalgia Impact Questionnaire (FIQ) subdomains (Function, Overall, and Symptoms) or total FIQ; for Multi Fatigue Inventory (MFI) subdomains (GF: general fatigue, Pfa: physical fatigue, RA: reduced activity, RM: reduced motivation, and MF: mental fatigue), and for Short-Form-36 Health Survey (SF-36) subdomains (PF: physical functioning, RP: role physical, BP: bodily pain, GH: general health, VT: vitality, SF: social functioning, RE: role emotional, and MH: mental health).

and the diagnosis received. Interestingly, activation of HERV and genes involved in immune response (Module 3) seem to strongly associate with diagnosis of ME/CFS (subgroup 2) ($|R| > 0.89$, $p < 0.001$) (*Figure 6A*). In fact, top correlated genes and HERV with ME/CFS diagnosis (*Figure 6B*), showed the strongest association with disease symptoms (*Figure 6C*), particularly with some domains of the MFI and SF-36 questionnaires, including physical and mental fatigue ($p < 0.05$). These results suggest the participation of these HERV and genes in ME/CFS disease while uncovering their potential as biomarkers.

## Discussion

Researchers have strived for decades to elucidate the etiology and pathophysiology of ME/CFS and FM, with disease complexity and patient heterogeneity hindering this task. With no biomarkers available and unclear pathophysiology, ME/CFS and FM are often misdiagnosed, further complicating the identification of effective treatments. In this study, we tried to overcome patient heterogeneity by studying a very finely phenotyped cohort diagnosed by a single expert clinician, representing one of the few studies (*Nepotchatykh et al., 2023*) including samples of patients diagnosed with either ME/CFS, FM, or both (co-diagnosed group) by two different clinical criteria (Canadian (*Carruthers et al., 2003*) and International Consensus (*Carruthers et al., 2011*) criteria, and the 1990 (*Wolfe et al., 1990*) and 2011 (*Wolfe et al., 2016*; *Wolfe et al., 2010*) ACR criteria, respectively). Due to previously reported sex-associated differences in these diseases, only female patients were included, limiting the external validity of the results in the male population while minimizing potential sex-associated bias.

Triggering agents for ME/CFS and FM, such as infections, stress, or trauma (*Chu et al., 2019*; *Furness et al., 2018*), seem capable of disrupting the epigenetic mechanisms constraining HERV expression (*Rangel et al., 2022*). Dysregulation of HERV has been proposed to contain more distinctive features than the host transcriptome itself in response to viral infections (*Marston et al., 2021*), and it is becoming detected in additional complex diseases (*Giménez-Orenga and Oltra, 2021*), offering a new perspective to study their physiopathology. In fact, HERV profiling could constitute an indirect approach (surrogate marker) to evidence epigenetic derangements and 3D alterations of the chromatin structure with obvious functional consequences. Previous research revealed aberrant activation of HERV in ME/CFS (*Rodrigues et al., 2019*) and FM (*Ovejero et al., 2020*). However, these two studies consisting of broad amplification of HERV by degenerate primer sets, did not allow for specific HERV loci identification. In this study, we fill this gap of knowledge by genome-wide analyzing annotated HERV sequences in the human genome and their relationship with immune gene expression and patient symptoms. The results show for the first time a comparison of PBMCs HERV expression profiles of ME/CFS, FM, and co-diagnosed patient groups as compared to HERV expression profiles of healthy subjects. Despite the limited number of samples (8 ME/CFS cases, 10 FM, 16 comorbid cases, and 9 matched healthy controls), our results show a specific HERV fingerprint for each disease that allows perfect discrimination into three distinct patient groups and their separation from healthy participants, opening the possibility of HERV expression profile-based diagnostic of these symptom-related, diseases. Methodology that could be extended to the study of other chronic diseases such as post-COVID-19 condition. Particularly interesting will be to determine whether post-COVID-19 cases present different HERV fingerprints that justify commonalities to ME/CFS and/or FM, or if by contrast show an evolution closer to the more prevalent co-diagnosed state. Worth noting are the results obtained in the co-diagnosed group which, unexpectedly, presented a completely different HERV profile to those identified for ME/CFS and for FM while strikingly closer to that of healthy controls. Findings that support a distinct subjacent pathomechanism for the co-diagnosed group of patients and, therefore, the identification of a novel nosologic entity.

In addition, our study reveals enhanced HERV dysregulation in the ME/CFS group when compared to FM, co-diagnosed cases, or healthy controls and significant quantitative differences within a more severely affected ME/CFS subgroup (subgroup 2) supporting inter-individual HERV expression variability toward patient subtyping, with potential relevance for patient tailored treatments, and potential for HERV expression levels to define degree of severity in ME/CFS. The study also defined a set of DE HERV commonly downregulated in all three patient groups (ME/CFS, FM, and their co-diagnosis, Cluster 4), a deeper study of which may help explain clinical commonalities between these conditions. In fact, the lack of a specific association of Cluster 4 DE HERV with a patient-study group opens the possibility of their derangement in additional complex and chronic diseases.

Analysis of DE HERV loci at family level within each cluster interestingly showed the involvement of different HERV families in each of the study groups, with ME/CFS outstanding by the largest heterogeneity, including deregulation in families such as HERV-H or HERV-P. Deregulation of HERV-H and HERV-P families has been previously linked to disease (*Bergallo et al., 2019*; *Morris et al., 2019*; *Yi et al., 2006*) with potential involvement in immune function performance (*Goodchild et al., 1993*; *Kulski et al., 1999*; *Mager et al., 1999*; *Mangeney et al., 2001*; *Matsuzawa et al., 2021*). Another HERV family consistently linked to disease is the HERV-W (*Dolei et al., 2014*; *Giménez-Orenga et al., 2022*; *Kremer et al., 2013*), whose upregulation has been reported in FM (*Ovejero et al., 2020*). Contrarily, our results evidenced downregulation of HERV-W family in all patient groups, particularly the HERV-W-LTRSU3_19q13.42_st element. Dissimilar results may be due to the use of different techniques to evaluate family expression (degenerate primers in RT-qPCR vs. probes in microarray), or accuracy in patient diagnosis. Our study further revealed that the majority of downregulated HERV in ME/CFS, FM, and co-diagnosed groups belonged to a sole HERV family, the MLT1. This family was found enriched near DE genes involved in the response to infection (*Bogdan et al., 2020*). MLT1 family members are among the most frequently overlapping with human lncRNAs (*Ramsay et al., 2017*). However, the potential role of MLT1-derived lncRNAs in the diseases studied here remains speculative at this stage.

In accordance with genomic data (*Lander et al., 2001*), most deregulated HERV loci consisted of solitary LTRs, known for their ability to influence gene expression through their regulatory sequences (*Ito et al., 2017*). For example, a HERV-H solitary LTR located at the *HHLA2* locus serves as its main polyadenylation signal (*Goodchild et al., 1993*; *Mager et al., 1999*), allowing the encoding of the HHLA2 protein, which regulates CD4 and CD8 T cell functions along with antigen-presenting cell proliferation (*Zhao et al., 2013*). Furthermore, it is known that non-allelic homologous recombination events leading to solitary LTRs create inter-individual variation, not only by removing coding potential but also by altering the *cis*-regulatory or transcriptional activity of the affected particular genomic regions (*Rebollo et al., 2012*; *Thomas et al., 2018*) even causing microdeletions (*Hermetz et al., 2012*; *Sanchez-Valle et al., 2010*; *Shuvarikov et al., 2013*), which offer possibilities for exploring familial risk factors of these diseases.

Analysis of differential gene expression in the exact same samples provided additional evidence of HERV-associated immune changes in the ME/CFS group as compared to FM, co-diagnosed, or healthy control groups, further supporting the existence of two ME/CFS subgroups differing, once more, at the quantitative level only.

Although genomic localization of correlated HERV and genes suggested major mechanisms other than co-transcription, it should be noted that gene probes included in the microarray do not fully picture the complete mRNA transcriptome, and therefore the possibility that other DE genes locate nearer to deregulated HERV cannot be ruled out. Furthermore, long-range interactions between gene promoters and LTR regulatory elements have been described for some HERV families, including MLT1, MER21, MER41, or LTR54 (*Ito et al., 2017*), reported as DE in this study. In this regard, it has been shown that the HERV-H family, upregulated in ME/CFS subgroup 2, can influence gene expression over long genomic distances, for example by creating topologically associated domains (*Zhang et al., 2019*), perhaps explaining at least some of our findings of DE immune genes being co-expressed with DE HERV. Under the hypothesis that independently transcribed DE HERV could be regulated by the same *trans*-acting mechanisms as DE genes, we interestingly found enrichment for TFBS regulating the immune response within DE HERV, suggesting *trans*-mediated coordination of deranged HERV and gene expression. Furthermore, HERV upregulated in ME/CFS were markedly enriched in chromatin remodeling factor binding sites, including SETDB1 and TRIM28, both being key epigenetic repressors of HERV expression, pointing to the impairment of TRIM28/SETDB1 axis as the underlying mechanism of HERV upregulation in ME/CFS. In support of this possibility, influenza A virus, a suspected trigger of ME/CFS (*Magnus et al., 2015*), has been shown to activate HERV expression by impairing the repressor TRIM28/SETDB1 axis (*Schmidt et al., 2019*).

As the literature endorses changes in immune cell subsets in ME/CFS (*Brenu et al., 2014*; *Brenu et al., 2011*; *Curriu et al., 2013*), we aimed to indirectly measure potentially skewed immune cell type ratios in our study groups with the use of CIBERSORTx software (*Newman et al., 2019*). The detected increase in plasma cells and resting CD4 T memory cells in ME/CFS subgroup 2, suggests the presence or the history of previous infections stimulating the immune system in these subjects, coinciding

with greater dysregulation in immune-related genes and HERV upregulation and increased severity of symptoms. On another hand, this same subgroup of patients exhibited lower levels of γδ T cells, an event reported to our knowledge in other autoimmune diseases like systemic lupus erythematosus and psoriasis (*Fay et al., 2016*; *Wang et al., 2012*). It has been described that γδ T cells mediate the immune response against viral infections like EBV, HHV-6, or CMV (*Zhao et al., 2018*), which together with natural killer cell dysfunction reported in the mentioned autoimmune diseases (*Henriques et al., 2013*; *Kim et al., 2019*) but also in ME/CFS (*Eaton-Fitch et al., 2019*), suggests an impaired cytotoxic function rendering patients with lower counts of this cell type, becoming more susceptible to viral reactivation. Interestingly, anomalies in immune cell abundance highly correlated with observed HERV expression patterns, indicating that HERV deregulation may either reflect expression derangement in those cell subpopulations or somehow influence immune cell ratios. The analysis of PBMC, rather than isolated subpopulations, is limited at distinguishing between these non-exclusive possibilities. In addition, we found strong associations between the HERV elements MER31_3p26.1 and HERV16_3p21.31 expression correlating with physical and mental fatigue according to health status assessment instruments MFI (*Smets et al., 1995*) and SF-36 (*Mchorney et al., 1993*), where higher scores indicate a worse or better health status, respectively. In this regard, both HERV elements, belonging to Module 3, that is, upregulated in ME/CFS subgroup 2, positively and negatively correlated with MFI and SF-36 subdomains, respectively, indicating that increased expression of MER31_3p26.1 and HERV16_3p21.31 elements may affect patient health status. Interestingly enough, previous correlations between HERV expression and physical fatigue are reported in the post-viral syndrome post-COVID-19 condition (*Giménez-Orenga et al., 2022*), which shares symptomatology with ME/CFS (*Komaroff and Lipkin, 2023*). Further study of MER31_3p26.1 and HERV16_3p21.31 elements and their potential pathogenic function seem granted.

Overall, this study not only shows the potential of HERV expression profiles as biomarkers for ME/CFS, FM, and for the definition of ME/CFS and FM co-diagnosis but also exposes their potential involvement in immune anomalies and connection with patient symptoms in these diseases, with etiological potential. Our results reveal family-specific HERV deregulation in ME/CFS and FM immune cells and their association with changes in pathogen-detection genes and immune cell subsets. Recent findings by *Marston et al., 2021* demonstrated specific disruption of HERV families in response to specific viral infection agents. Perhaps a database enabling the comparison of disease HERV expression profiles with those induced by exogenous infections could help reveal the triggering agents behind diseases of suspected viral etiology, such as ME/CFS and FM.

In summary, this study pioneers the comparison of female ME/CFS, FM, and co-diagnosed HERV expression profiles showing the superior diagnostic potential of HERV fingerprints over immune transcriptomes, with clinical implications, importantly providing an effective tool for the objective differential diagnosis of these hard-to-diagnose co-occurring diseases. Given the limited sample size of our cohort, validation of the findings in extended cohorts is a must. The potential mechanisms point at underlying epigenetic disturbances strikingly derepressing HERV silencing, particularly in ME/CFS. Whether these disturbances primarily affect certain immune subpopulations, estimated decreased for γδ T cells, or increased for plasma and resting CD4 memory T cells, correlating with patient symptom severity in ME/CFS, or reflect the presence of DEHERV profiles in immune cell subpopulations biased in these patient groups is an aspect that awaits further work.

## Materials and methods
### Study design
This cross-sectional observational study was approved by the Public Health Research Ethics Committee DGSP-CSISP of Valencia, núm. 20190301/12, Valencia, Spain. The study included a total of 43 female patients invited to participate from local patient associations who were clinically diagnosed with ME/CFS (*n* = 8), FM (*n* = 10), or both (*n* = 16) (National Biobank Registry Ref 0006024) and 9 healthy control individuals' population-matched for age and BMI (National Biobank Registry Ref 0006034). Patients were diagnosed by an FM and ME/CFS specialized clinician (Hospital de Manises, Valencia, Spain) using the 1990 (*Wolfe et al., 1990*) and 2011 (*Wolfe et al., 2016*; *Wolfe et al., 2010*) American College of Rheumatology (ACR) criteria for FM and the Canadian (*Carruthers et al., 2003*) and International Consensus (*Carruthers et al., 2011*) criteria for ME/CFS diagnosis. Patients with health

problems other than FM and ME/CFS were excluded from the study. Individuals with any similar o related pathology, including a medical history of chronic pain and/or fatigue, or serious health complications, were excluded from control group, as well as medicated healthy controls. Written informed consent was obtained from all study participants and patient health status was also evaluated with the use of standardized questionnaires, including the FIQ case report form (*Burckhardt et al., 1991*; *Rivera and González, 2004*), the MFI questionnaire (*Smets et al., 1995*), and the quality-of-life SF-36 instrument (*Mchorney et al., 1993*). Participating patients agreed to withdraw medication at least 12 hr prior to blood draw. Microarray analysis of RNA extracted from PBMCs was performed along with complementary bioinformatic analysis to evaluate differences in HERV profiles among patients with potential application as biomarkers or indicators of the underlying pathomechanisms.

## Isolation of PBMCs and total RNA extraction

Up to 10 ml of whole blood were collected via venipuncture after a 12-hr overnight fasting in K2EDTA tubes (Becton Dickinson, Franklin Lakes, NJ, USA) and processed within 2 hr by dilution at 1:1 (vol/vol) ratio in phosphate-buffered saline solution (PBS) with layering on top of 1 volume of Ficoll-Paque Premium (GE Healthcare, Chicago, IL, USA) and separation by density centrifugation at $500 \times g$ for 30 min (20°C, brakes off). The PBMC layer was isolated and washed with PBS and resuspended in red blood cell lysis buffer (155 mM $NH_4Cl$, 10 mM $NaHCO_3$, 0.1 mM EDTA, and pH 7.4), kept on ice for 5 min, and centrifuged (20°C at $500 \times g$ for 10 min), to remove contaminating erythrocytes. The washed pellets were adjusted to a final concentration of $10^7$ cells/ml in freezing medium (90% FBS, 10% DMSO), aliquoted, and deeply frozen in liquid nitrogen until use. Total RNA was extracted using RNeasy Mini Kit (QIAGEN, MD, USA) according to the manufacturer's instructions. RNA quality was assessed using Agilent TapeStation 4200 (Agilent). All RNA samples had an RNA integrity number above 7.

## Transcriptome analysis by microarray

To minimize batch effects the samples were anonymized (blinded to the operator) and scrambled across groups. HERV transcriptome was scrutinized using custom high-density HERV-V3 microarrays (*Becker et al., 2017*), capable to discriminate 174,852 HERV elements, 179,142 MaLR elements, and putative active 1072 LINE-1 elements at the locus level, in addition to detecting a set of 1559 genes involved in eight potentially relevant cellular pathways (immunity, inflammation, cancer, central nervous system affections, differentiation, telomere maintenance, chromatin structure, and gag-like genes). These custom HERV-V3 arrays also allow to discern the different subdomains within proviral sequences: 3′ or 5′ LTRs and *gag/pol/env* regions of proviral HERV and solitary LTRs originated after non-allelic recombination events between the two LTRs causing complete proviral elimination (*Thomas et al., 2018*). To analyze HERV transcriptomes of PBMC-derived RNA samples, cDNA was synthesized and amplified from 45 ng of RNA using the Ovation Pico WTA System V2 kit (Nugen) according to the manufacturer's instructions. The resulting amplified ssDNA was purified using the QIAquick purification kit (QIAGEN, MD, USA). Total DNA concentration was measured with NanoDrop 2000 spectrophotometer (Thermo Scientific) and quality was assessed on the Bioanalyzer 2100. Five micrograms of purified ssDNA were enzymatically fragmented into 50–100 bp fragments and biotin-labeled with the Encore Biotin Module kit from Nugen according to the manufacturer's instructions. The resulting target was mixed with standard hybridization controls and B2 oligonucleotides following supplier recommendations. The hybridization cocktail was heat-denatured at 95°C for 2 min, incubated at 50°C for 5 min, and centrifuged at $16,000 \times g$ for 5 min to pellet the residual salts. HERV-V3 microarrays were pre-hybridized with 200 µl of hybridization buffer and placed under stirring (60 rpm) in an oven at 50°C for 10 min. Hybridization buffer was then replaced by denatured hybridization cocktail. Hybridization was performed at 50°C for 18 hr in the oven under constant stirring (60 rpm). Washing and staining were carried out according to the manufacturer's protocol, using a fluidic station (GeneChip fluidic station 450, Affymetrix). Arrays were finally scanned using a fluorometric scanner (GeneChip scanner 3000 7G, Affymetrix). Biotin purification, hybridization, and reading steps were performed by Sampled (Piscataway, NJ 08854, USA). Overall, 1,397,352 probes were detected, the vast majority of them corresponding to HERV (1,290,800 probesets), followed by genes (103,724 probesets), and LINE-1 (2828 probesets).

### Identification of DE HERV and genes

All bioinformatic analyses were performed with RStudio software version 4.2.1. Microarray CEL files were processed and analyzed using R oligo package (*Carvalho and Irizarry, 2010*). Data were normalized, adjusted for background noise, and summarized using the RMA (Robust Multi-Array) algorithm. DE analysis was performed using limma R package (*Ritchie et al., 2015*), considering DE those probes with a Benjamini–Hochberg adjusted p-value (FDR) <0.1 and an absolute $\log_2$ fold-change >1. The analysis results were presented in volcano plots using EnhancedVolcano R package (*Blighe et al., 2022*).

### Clustering analysis

DE HERV or genes (FDR <0.1 and $|\log_2 FC| > 1$) were used to perform clustering analysis. PCA was performed to analyze the behavior of samples under study based on DE HERV expression profiles by using factoextra R package (*Kassambara and Mundt, 2025*). Samples were plotted according to the first two PCA principal components and the contribution of variables to each of the principal components was represented in barplots. Scaled expression data of DE HERV or gene probesets were used to cluster samples and probes. Data were represented in heatmap plots by using pheatmap R package (*Kolde et al., 2018*). Rows and columns were clustered using Euclidean and Pearson correlation distances, respectively. HERV clusters were visually identified based on heatmap results and extracted from dendrograms using the pheatmap implemented function. Mean probe expression by group was calculated for all probes encompassed in each cluster and their distribution and expression level were visualized by box and violin plots. Average-linkage agglomerative hierarchical clustering with Pearson correlation distance was performed on normalized expression data to identify modules of HERV and genes with correlated expression patterns across samples. Modules of HERV–genes were produced by applying a dynamic tree cut of dendrograms using a minimum cluster size of 110 HERV or genes, as described by *Marston et al., 2021*.

### Enrichment analysis

Microarray probes were annotated based on the HERV-V3 annotation file kindly provided by bioMérieux, allowing retrieval of HERV subdomain (gag, pol, env, or LTR), family, and genomic location of each probe. Family group information for solitary LTR probes was added based on Dfam and EnHERV databases. Original hg37 HERV-V3 coordinates were converted into hg38 genomic coordinates by using UCSC LiftOver tool (*Kuhn et al., 2013*) and the corresponding chromosome band information was added. The genomic context of the HERV loci was assessed with Goldmine R package (*Bhasin and Ting, 2016*). Enrichment analysis for HERV family, HERV subdomain, and genomic context was performed for the HERV encompassed in each cluster by comparing their frequency in the cluster to their corresponding on hg38 genomic data according to HERV-V3 microarray annotation (*Becker et al., 2017*). The frequency of families, HERV subdomains, and genomic context (intron, exon, intergenic, promoter, or 3′ end) for the DE HERV encompassed in each cluster and in the overall hg38 genomic data were represented as stacked bar plots with ggplot2 R package (*Wickham, 2016*). To assess for TFBS enrichment, the ReMapEnrich Shiny 1.4 web interface was used providing DE HERV loci genomic coordinates as input. Enriched TFBS were represented as a heatmap using pheatmap R package (*Kolde et al., 2018*), with colors defining modules in which the enrichment was found, and no enrichments were shown as blank.

### Overrepresentation functional analysis

Genes encompassed in modules identified by hierarchical clustering as well as TF binding to identified enriched TFBS in HERV of each module were functionally analyzed by GO using the R package clusterProfiler (*Wu et al., 2021*) with an adjusted p-value cutoff of 0.05. For each module, the top 5 GO terms as ranked by increasing adjusted p-value were visualized in a heatmap using pheatmap R package (*Kolde et al., 2018*). Statistical significance of each GO term as adjusted p-values was represented by a color gradient, with darker colors representing more significant terms in the module. Terms sharing significant overrepresentation in other modules (adjusted p < 0.05) were colored on heatmaps accordingly.

### Evaluation of overlaps between HERV and gene sequences

Genomic coordinates of HERV and genes encompassed in each module were transformed to genomic range objects (Granges) using the GenomicRanges R package (*Lawrence et al., 2013*). They were

subsequently plotted over a linear chromosomal representation with karyoploteR package (*Gel and Serra, 2017*), which displays the density of sequences throughout chromosomes. The genomic distance between HERV and genes in the same module was determined with a specific function included in GenomicRanges package and displayed as density plots.

### Measurement of immune cell relative amounts

Digital cytometry analysis was performed with CIBERSORTx (*Newman et al., 2019*) software on normalized non-logarithmic gene expression data for all samples. The default LM22 leukocyte gene signature matrix (*Newman et al., 2015*) was used as a reference. LM22 was generated using Affymetrix HGU133A microarray data distinguishing 22 human hematopoietic cell phenotypes, including different types of T cells, B cells (naïve and memory), plasma cells, NK cells, monocytes, dendritic cells, and some myeloid subsets (*Newman et al., 2015*). Output results were visually inspected for potential differences in immune cell counts and significant differences were assessed by Welch's *t*-test with adjusted p-value <0.05 from specific immune cell types. Results were plotted as violin plots.

### Trait-measure and symptom association

For each module extracted by hierarchical clustering the eigengenes were calculated with the use of the function moduleEigengenes implemented in WGCNA package (*Langfelder and Horvath, 2008*). A matrix of normalized expression data and a list assigning each sequence to each module was provided as input. Calculated module eigengenes were then correlated to external microarray data, including immune cell proportion values provided by CIBERSORTx, questionnaire scores for FIQ, MFI, and SF-36, or the clinical diagnosis. This last variable was binarized with the use of binarizeCategorialColumns. Results were plotted by using pheatmap package (*Kolde et al., 2018*) with boxes showing Pearson correlation values and associated p-values between gene expression levels of each module and the disease trait. A correlation value of 1 (orange or green) and –1 (blue) were used to quantify the strongest positive and negative correlations, respectively, while 0 (white) was assigned for no correlation.

### Real-time quantitative polymerase chain reaction

DE of specific HERV sequences was validated using the same RNA samples analyzed by microarray (*n* = 8 controls, *n* = 8 ME/CFS, *n* = 7 FM, and *n* = 14 co-diagnosed). Reverse transcription was performed using High-Capacity cDNA reverse Transcription kit (Applied Biosystems, Waltham, MA, USA, cat. 4308228), with 1 µg of total RNA according to the manufacturer's guidelines. cDNAs were used for real-time PCR using PowerUP Sybr Green Master Mix (Applied Biosystems, cat. 100029283) and a Lightcycler LC480 instrument (Roche, Penzberg, Germany). Standard amplification conditions were applied, including a single hotstart polymerase preactivation cycle at 94°C for 15 min, up to 45 amplification cycles, each one consisting of three steps: denaturation at 95°C for 15 s, annealing at 60°C for 30 s, and extension at 70°C for 30 s. Sequences of specific primers used are detailed in *Supplementary file 1H*. GAPDH levels were used for relative quantification of the RNAs amplified, and 2−ΔCt analysis to calculate expression was applied.

### Statistical analysis

All statistical analyses were done in R v4.2.1. Data distributions were tested for normality. Normally distributed data were tested using two-tailed unpaired Student's *t*-tests; non-normal data were analyzed with non-parametric statistical test, as detailed. For enrichment analyses, we used a Fisher's exact test to calculate p-value, considering enriched in the provided list if an adjusted p-value (FDR) was less than 0.05. We chose to use Fisher's exact test in the analysis of contingency tables to compare ME/CFS and healthy group, as it is more appropriate for small sample sizes in comparison to the chi-square test or *G*-test of independence.

## Acknowledgements

We particularly thank all the patients who participated in the study and Dr. Vicente Serra (Umivale, Valencia, Spain) for his help in the recruitment of healthy volunteers. We also acknowledge bioMérieux (Ain, France) for allowing the use of their custom-made HERV-V3 high-density microarrays and Sampled LTD (Piscataway, NJ, USA) for assisting us in the analysis by microarray. This study was funded by an ME

Research UK (SCIO charity number SC036942) grant, and by Generalitat Valenciana CIAICO/2021/103 grant to EO. KG-O is supported by the Generalitat Valenciana ACIF2021/179 grant. Funders were not involved in any of the research stages.

## Additional information

### Funding

| Funder | Grant reference number | Author |
| --- | --- | --- |
| ME Research UK | SC036942 | Elisa Oltra |
| Generalitat Valenciana | CIAICO/2021/103 | Elisa Oltra |
| Generalitat Valenciana | ACIF2021/179 | Karen Giménez-Orenga |

The funders had no role in study design, data collection, and interpretation, or the decision to submit the work for publication.

### Author contributions

Karen Giménez-Orenga, Data curation, Formal analysis, Investigation, Writing – original draft, Writing – review and editing; Eva Martín-Martínez, Investigation, Methodology, Writing – review and editing; Lubov Nathanson, Investigation, Writing – review and editing; Elisa Oltra, Conceptualization, Data curation, Supervision, Funding acquisition, Writing – original draft, Project administration, Writing – review and editing

### Author ORCIDs

Karen Giménez-Orenga ⓘ https://orcid.org/0000-0001-9790-7327
Eva Martín-Martínez ⓘ https://orcid.org/0000-0003-3261-494X
Lubov Nathanson ⓘ https://orcid.org/0000-0003-1038-9083
Elisa Oltra ⓘ https://orcid.org/0000-0003-0598-2907

### Ethics

This study has been performed in accordance with the Declaration of Helsinki. It was approved by the Public Health Research Ethics Committee DGSP-CSISP of Valencia, núm. 20190301/12, Valencia, Spain. All participants signed an informed consent to participate in this study.

Reviewer #1 (Public review): https://doi.org/10.7554/eLife.104441.3.sa1
Reviewer #2 (Public review): https://doi.org/10.7554/eLife.104441.3.sa2
Reviewer #3 (Public review): https://doi.org/10.7554/eLife.104441.3.sa3
Author response https://doi.org/10.7554/eLife.104441.3.sa4

## Additional files

### Supplementary files

MDAR checklist

Supplementary file 1. Raw, processed, and complementary data. (**A**) Itemized participant demographics and health status assessment with FIQ, MFI and SF-36 questionnaires. (**B**) Statistical analysis of participant demographics and health status assessment with FIQ, MFI and SF-36 questionnaires. (**C**) Differentially expressed HERV. (**D**) Patient health status assessment with FIQ, MFI, and SF-36 questionnaires separating ME/CFS into two subgroups. (**E**) Relative contribution of HERV families to each cluster. (**F**) Differentially expressed genes. (**G**) Genomic distance between correlated HERVs and genes in each module. (**H**) Primers for RT-qPCR validation.

### Data availability

All data associated with the study are available in the paper or supplementary materials, except for HERV-V3 microarray annotation provided by bioMérieux under material transfer agreement. Microarray data is available at GSE269047 from the NCBI GEO database.

The following dataset was generated:

| Author(s) | Year | Dataset title | Dataset URL | Database and Identifier |
|---|---|---|---|---|
| Giménez-Orenga K, Martín-Martínez E, Nathanson L, Oltra E | 2025 | HERV activation segregates ME/CFS from fibromyalgia while defining a novel nosologic entity | https://www.ncbi.nlm.nih.gov/geo/query/acc.cgi?acc=GSE269047 | NCBI Gene Expression Omnibus, GSE269047 |

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
