## [Editor Report · eLife Assessment]

This **important** study substantially expands observations of HERV expression in the clinical settings. The evidence provided by the authors that HERV activity is an underlying etiological factor in ME/CFS and fibromyalgia is **compelling** and suggests further investigation into mechanisms. This work will be of broad interest to clinicians and researchers alike.

---

## [Referee Report · Reviewer #1 (Public review)]

Summary:

Giménez-Orenga et al. investigate the origin and pathophysiology of myalgic encephalomyelitis/chronic fatigue syndrome (ME/CFS) and fibromyalgia (FM). Using RNA microarrays, the authors compare the expression profiles and evaluate the biomarker potential of human endogenous retroviruses (HERV) in these two conditions. Altogether, the authors show that HERV expression is distinct between ME/CFS and FM patients, and HERV dysregulation is associated with higher symptom intensity in ME/CFS. HERV expression in ME/CFS patients is associated with impaired immune function and higher estimated levels of plasma cells and resting CD4 memory T cells. This work provides interesting insights into the pathophysiology of ME/CFS and FM, creating opportunities for several follow-up studies.

Strengths:

(1) Overall, the data is convincing and supports the authors' claims. The manuscript is clear and easy to understand, and the methods are generally well-detailed. It was quite enjoyable to read.

(2) The authors combined several unbiased approaches to analyse HERV expression in ME/CFS and FM. The tools, thresholds, and statistical models used all seem appropriate to answer their biological questions.

(3) The authors propose an interesting alternative to diagnosing these two conditions. Transcriptomic analysis of blood samples using an RNA microarray could allow a minimally invasive and reproducible way of diagnosing ME/CFS and FM.

Weakness:

(1) While this work makes several intriguing observations, some results will need to be validated in future studies using experimental approaches.

---

## [Referee Report · Reviewer #2 (Public review)]

Summary:

Giménez-Orenga carried out this study to assess whether human endogenous retroviruses (HERVs) could be used to improve the diagnosis of Myalgic Encephalomyelitis/Chronic Fatigue Syndrome (ME/CFS) and Fibromyalgia (FM). To this end, they used the HERV-V3 array developed previously, to characterize the genome-wide changes in expression of HERVs in patients suffering from ME/CFS, FM or both, compared to controls. In turn, they present a useful repertoire of HERVs that might characterize ME/CFS and FM. For most part, the paper is written in a manner that allows a natural understanding of the workflow and analyses carried out, making it compelling. The figures and additional tables presents solid support for the findings. However, some statements made by the authors seem incomplete and would benefit by a more thorough literature review. Overall, this work will be of interest to the medical community seeking in better understanding the co-occurrence of these pathologies, hinting at a novel angle by integrating HERVs, which are often overlooked, into their assessment.

Strengths:

- The work is well-presented, allowing the reader to understand the overall workflow and how the specific aims contribute to filling the knowledge gap in the field.

- The analyses carried out to understand the potential impact on gene expression mediated by HERVs are in line with previous works, making it solid and robust in the context of this study.

Weaknesses:

- The authors claim to obtain genome-wide HERV expression profiles. However, the array used was developed using hg19, while the genomic analysis of this work are carried out using a liftover to hg38. It would improve the statement and findings to include a comparation of the differences in HERVs available in hg38, and how this could impact the "genome-wide" findings.

- The authors in some points are not thorough with the cited literature. Two examples are:

(1) Lines 396-397 the authors say "the MLT1, usually found enriched near DE genes (Bogdan et al., 2020)". I checked the work by Bogdan, and they studied bacterial infection. A single work in a specific topic is not sufficient to support the statement that MLT1 is "usually" in close vicinity to differentially expressed genes. More works are needed to support this.

(2) After the previous statement, the authors go on to mention "contributing to the coding of conserved lncRNAs (Ramsay et al., 2017)". First, lnc = long non-coding, so this doesn't make sense. Second, in the work by Ramsay they mention "that contributed a significant amount of sequence to primate lncRNAs whose expression was conserved", which is different to what the authors in this study are trying to convey. Again, additional work and a rephrasing might help to support this idea.

- When presenting the clusters, the authors overlook the fact that cluster 4 is clearly control-specific, and fail to discuss what this means. Could this subset of HERV be used as bona fide markers of healthy individuals in the context of these diseases? Are they associated with DE genes? What could be the impact of such associations?

Appraisals on aims:

The authors set specific questions and presented the results to successfully answer them. The evidence is solid, with some weaknesses discussed above that will methodologically strengthen the work.

Likely impact of work on the field:

This work will be of interest to the medical community looking for novel ways to improve clinical diagnosis. Although future works with a greater population size, and more robust techniques such as RNA-Seq, are needed, this is the first step in presenting a novel way to distinguish these pathologies.

It would be of great benefit to the community to provide a table/spreadsheet indicating the specific genomic locations of the HERVs specific to each condition. This will allow proper provenance for future researchers interesting in expanding on this knowledge, as these genomic coordinates will be independent of the technique used (as was the array used here).

Comments on revisions:

When addressing the comments made in the previous round, there are some answers that lack substance and don't seem to be incorporated in the manuscript. For example, the authors say:

Authors' response: This is an important point. However, the low number of probes (less than 100) that were excluded from our analysis by lack of correspondence with hg38 among the 1,290,800 probesets was interpreted as insignificant for "genome-wide" claims. An aspect that will be explained in the revised version of this manuscript.

I checked the revised manuscript with tracked changes, and there doesn't seem to be an updated explanation to this. In which lines is this explained?

For the other response:

Authors' response: Using control DE HERV as bona fide markers of healthy individuals seems like an interesting possibility worth exploring. Control DE HERV (cluster 4) associate with DE genes involved in apoptosis, T cell activation and cell-cell adhesion (modules 1 and 6). The impact of which deserves further study.

I couldn't find an updated mention of this in the discussion.

Another point that I raised was regarding the decision of using an FDR of 0.1 instead of 0.05. The authors only speculate about the impacts in their answer, while I believe that this could have been rigorously addressed. Since this was done in R, and DE analysis are relatively fast, I don't see a reason as to why this part was not repeated and discussed accordingly.

For other analyses, there doesn't seem to be a problem with using 0.05 as threshold. Examples of this are the "Overrepresentation functional analysis", or the "Statistical analysis" part of the methods they say "we used a Fisher exact test to calculate p-value, considering enriched in the provided list if an adjusted p-value (FDR) was less than 0.05".

Just to make this point clear: I'm not asking the authors to repeat all the work using the 0.05 FDR threshold, but rather that they are aware and conscious about the impact of this, and give an idea to the audience on how it would change the DE numbers. This would put in perspective the findings to any future reader.

I think that most of the other answers to both my previous concerns and the other reviewer's concerns are ok. My last outstanding concern is that the probe coordinates apparently can't be shared, which undermines a lot this study reproducibility, and its use by future researches which won't be able to compare their results to this study.

---

## [Referee Report · Reviewer #3 (Public review)]

Summary:

The authors find that HERV expression patterns can be used as new criteria for differential diagnosis of FM and ME/CFS and patient subtyping. The data are based on transcriptome analysis by microarray for HERVs using patient blood samples, followed by differential expression of ERVs and bioinformatic analyses. This is a standard and solid data processing pipeline, and the results are well presented and support the authors' claim.

Strengths:

It provides an innovative diagnostic approach using ERV profiles to subtype patients and distinguish FM and ME/CFS.

Comments on revisions:

This is a revised manuscript which addresses the comments well.

---

## [Author Response]

The following is the authors’ response to the original reviews

**Public Reviews:**

**Reviewer #1 (Public review):**
Summary:Giménez-Orenga et al. investigate the origin and pathophysiology of myalgic encephalomyelitis/chronic fatigue syndrome (ME/CFS) and fibromyalgia (FM). Using RNA microarrays, the authors compare the expression profiles and evaluate the biomarker potential of human endogenous retroviruses (HERV) in these two conditions. Altogether, the authors show that HERV expression is distinct between ME/CFS and FM patients, and HERV dysregulation is associated with higher symptom intensity in ME/CFS. HERV expression in ME/CFS patients is associated with impaired immune function and higher estimated levels of plasma cells and resting CD4 memory T cells. This work provides interesting insights into the pathophysiology of ME/CFS and FM, creating opportunities for several follow-up studies.Strengths:(1) Overall, the data is convincing and supports the authors' claims. The manuscript is clear and easy to understand, and the methods are generally well-detailed. It was quite enjoyable to read.(2) The authors combined several unbiased approaches to analyse HERV expression in ME/CFS and FM. The tools, thresholds, and statistical models used all seem appropriate to answer their biological questions.(3) The authors propose an interesting alternative to diagnosing these two conditions. Transcriptomic analysis of blood samples using an RNA microarray could allow a minimally invasive and reproducible way of diagnosing ME/CFS and FM.Weaknesses:(1) The cohort analysed in this study was phenotyped by a single clinician. As ME/CFS and FM are diagnosed based on unspecific symptoms and are frequently misdiagnosed, this raises the question of whether the results can be generalised to external cohorts.

Thank you for your comment. Surely the study of larger cohorts will determine the external validity of these results in a clinical scenario. However, this pilot study, first of its kind, was designed to maximize homogeneity across participants which seemed primarily ensured by the study of females only and diagnosis by a single experienced observer.

(2) The analyses performed to unravel the causes and effects of HERV expression in ME/CFS and FM are solely based on sequencing data. Experimental approaches could be used to validate some of the transcriptomic observations.

Certainly, experimental approaches may add robustness to the implication of HERVs in ME/CFS. We indeed consider taking this avenue to deepen in the findings presented here for future work. However, the limited knowledge of HERV-mediated physiological functions may hamper the obtention of prompt results towards revealing causes and effects of HERV expression in ME/CFS and FM.

**Reviewer #2 (Public review):**
Summary:Giménez-Orenga carried out this study to assess whether human endogenous retroviruses (HERVs) could be used to improve the diagnosis of Myalgic Encephalomyelitis/Chronic Fatigue Syndrome (ME/CFS) and Fibromyalgia (FM). To this end, they used the HERV-V3 array developed previously, to characterize the genome-wide changes in the expression of HERVs in patients suffering from ME/CFS, FM, or both, compared to controls. In turn, they present a useful repertoire of HERVs that might characterize ME/CFS and FM. For the most part, the paper is written in a manner that allows a natural understanding of the workflow and analyses carried out, making it compelling. The figures and additional tables present solid support for the findings. However, some statements made by the authors seem incomplete and would benefit from a more thorough literature review. Overall, this work will be of interest to the medical community seeking in better understanding of the co-occurrence of these pathologies, hinting at a novel angle by integrating HERVs, which are often overlooked, into their assessment.Strengths:(1) The work is well-presented, allowing the reader to understand the overall workflow and how the specific aims contribute to filling the knowledge gap in the field.(2) The analyses carried out to understand the potential impact on gene expression mediated by HERVs are in line with previous works, making it solid and robust in the context of this study.Weaknesses:(1) The authors claim to obtain genome-wide HERV expression profiles. However, the array used was developed using hg19, while the genomic analysis of this work are carried out using a liftover to hg38. It would improve the statement and findings to include a comparison of the differences in HERVs available in hg38, and how this could impact the "genome-wide" findings.

This is an important point. However, the low number of probes (less than 100) that were excluded from our analysis by lack of correspondence with hg38 among the 1,290,800 probesets was interpreted as insignificant for "genome-wide" claims. An aspect that will be explained in the revised version of this manuscript.

(2) The authors in some points are not thorough with the cited literature. Two examples are:a) Lines 396-397 the authors say "the MLT1, usually found enriched near DE genes (Bogdan et al., 2020)". I checked the work by Bogdan, and they studied bacterial infection. A single work in a specific topic is not sufficient to support the statement that MLT1 is "usually" in close vicinity to differentially expressed genes. More works are needed to support this.b) After the previous statement, the authors go on to mention "contributing to the coding of conserved lncRNAs (Ramsay et al., 2017)". First, lnc = long non-coding, so this doesn't make sense. Second, in the work by Ramsay they mention "that contributed a significant amount of sequence to primate lncRNAs whose expression was conserved", which is different from what the authors in this study are trying to convey. Again, additional work and a rephrasing might help to support this idea.

Certainly, these two sentences need rephrasing to better adjust to current evidence.

Revised sentences can now be found in lines 397-402

(3) When presenting the clusters, the authors overlook the fact that cluster 4 is clearly control-specific, and fail to discuss what this means. Could this subset of HERV be used as bona fide markers of healthy individuals in the context of these diseases? Are they associated with DE genes? What could be the impact of such associations?

Using control DE HERV as bona fide markers of healthy individuals seems like an interesting possibility worth exploring. Control DE HERV (cluster 4) associate with DE genes involved in apoptosis, T cell activation and cell-cell adhesion (modules 1 and 6). The impact of which deserves further study.

Appraisals on aims:The authors set specific questions and presented the results to successfully answer them. The evidence is solid, with some weaknesses discussed above that will methodologically strengthen the work.Likely impact of work on the field:This work will be of interest to the medical community looking for novel ways to improve clinical diagnosis. Although future works with a greater population size, and more robust techniques such as RNA-Seq, are needed, this is the first step in presenting a novel way to distinguish these pathologies.It would be of great benefit to the community to provide a table/spreadsheet indicating the specific genomic locations of the HERVs specific to each condition. This will allow proper provenance for future researchers interested in expanding on this knowledge, as these genomic coordinates will be independent of the technique used (as was the array used here).

We agree with the reviewer that sharing genomic locations of DE HERVs in these pathologies would contribute to the development of these findings. Unfortunately, we do not hold the rights to share probe coordinates from this custom HERV-V3 microarray which we used under MTA agreement with its developer.

**Reviewer #3 (Public review):**
The authors find that HERV expression patterns can be used as new criteria for differential diagnosis of FM and ME/CFS and patient subtyping. The data are based on transcriptome analysis by microarray for HERVs using patient blood samples, followed by differential expression of ERVs and bioinformatic analyses. This is a standard and solid data processing pipeline, and the results are well presented and support the authors' claim.
**Recommendations for the authors:**

**Reviewer #1 (Recommendations for the authors):**
Recommandations/questions:(1) The authors point towards the biomarker potential of HERV expression signatures. In line with this, it would be important to test if they can predict the correct pathology for patients using the expression of DE HERVs. Additionally, as a single clinician annotated the cohort analysed in this study, it would be interesting to validate the signatures identified in this work by reanalysing publicly available transcriptomic data from independent studies.

Thank you for the suggestion. We plan to conduct this analysis and have added the following statement to the manuscript (lines 482-483): “Given the limited sample size in our cohort, validation of the findings in extended cohorts is a must.”

(2) The authors suggest that an epigenetic mechanism causes the dysregulated HERV expression in ME/CFS patients. However, in Fig.1A, HERV expression profiles of co-diagnosed patients are more similar to healthy controls than patients with either condition. How could the co-morbidity of FM "rescue" the phenotype of ME/CFS?

Thank you for the insightful comment. It is notable that co-diagnosed patients exhibit HERV expression profiles more similar to those of healthy controls than to either FM´s or ME/CFS´s. These findings may suggest a distinct underlying pathomechanism for this patient group, supporting the identification of a novel nosologic entity, as discussed in lines 372-374 of the manuscript.

(3) Abundant evidence in the literature links HERV dysregulation with the production of RNA:DNA hybrids and dsRNAs and viral mimicry. The authors found that ME/CFS subgroup 2, which exhibits the most important HERV dysregulation, is also associated with decreased signatures of pathogen detection. It would be interesting to quantify the abundance of DNA:RNA hybrids and dsRNAs in PBMCs of ME/CFS and FM patients as well as healthy controls. It would be interesting to discuss how downregulation of pathogen detection pathways could be a mechanism in ME/CFS patients to avoid viral mimicry and potential links with inflammation in this disease.

Certainly, HERVs can influence disease pathophysiology by generating RNA:DNA hybrids and dsRNA. However, microarray data does not allow this analysis. Future actions to investigate the underlying mechanisms of differentially expressed HERVs could investigate this interesting possibility.

(4) Another intriguing result is how overexpression of Module 3 in ME/CFS subgroup 2 is associated with higher levels of plasma cells. The authors hypothesize that the changes in immune cell abundances reflect previous viral infections, but another possibility would be immune activation against HERVs. Are there protein-coding sequences (gag, pro, pol, env) amongst the HERV sequences of module 3? If so, it would be interesting to validate HERV protein expression in these samples. Additionally, blood samples of ME/CFS patients and healthy controls should be analysed in flow cytometry to describe the abundance and phenotype of immune cells precisely.

Thank you for your insightful comments. In fact, we identified three HERV elements with protein-coding regions whose functional relevance remains uncertain. They present an interesting avenue for future investigation, particularly regarding immune activation.

Minor comments:(1) On lines 170-172, it is unclear to me how Figure 1E is linked to the text.

We have added a line better explaining Fig. 1E: “Top 10 contributing HERVs to principal components PC1 and PC2 are shown” (lines 171-172).

(2) Figure S2: grouping or colouring the plots based on the cluster to which HERVs were assigned could facilitate the understanding of the figure.

We appreciate the suggestion to enhance the clarity of the figures. However, this color-coding cannot be implemented, as a family is not exclusively assigned to a single cluster.

(3) How are the 4 HERV clusters of Figure 2 and the 8 modules of Figure 3 related to the clusters identified by hierarchical clustering in Figure 1? More details should be provided in the text (Results and Methods sections), and figures to illustrate the clustering strategy should be added if needed.

To enhance clarity, we have included the following explanation in the results section (lines 244-251): “To uncover potentially affected physiologic functions linked to DE HERV, we examined how DE HERVs and DE genes with similar expression patterns grouped together in modules based on their intrinsic relationships by their hierarchical co-clustering (Fig. 3). Then, the functional significance of these modules was assessed by gene ontology (GO) analysis of the DE genes within each module. The hierarchical clustering analysis resulted in the identification of eight distinct modules, each characterized by unique combinations of DE HERV and DE gene patterns across all four study groups (Fig. 3)”.

(4) Related to Figure 4, are there HERV sequences in module 3 located near genes important for plasma cells and/or resting CD4 memory T cells?

Thank you for your insightful comment. However, gene relevance for plasma cells and/or resting CD4 memory T cells may depend on multiple factors in addition to cell type and subtypes and, therefore, the analysis may not be straight forward.

**Reviewer #2 (Recommendations for the authors):**
In Figure 1, the heatmap scale goes from -4 to 4. This should reflect at least the numbers on the lowest and highest end of the scale.

Thank you for bringing this to our attention. The scale was correct; however, when arranging the panels, the numbers were not properly positioned. The figure has now been updated with the corrected version.

Figure 2F and G, percentages are shown as decimal numbers up to 1.00, while it should be 100%, and so on.

We also replaced this figure, changing the numbers to fit percentages.

It would be interesting to know how the results change using FDR of 0.05. I'm not familiar with microarray thresholds, but in RNA-Seq, 0.1 is rarely used, with 0.05 being the standard. Could it be that a more stringent result better distinguishes the pathologies?

Applying a more stringent threshold, such as FDR 0.05, may remove sequences that, while not strongly differentially expressed, may be still important for distinguishing between these pathologies. Therefore, we decided to also include DE tendencies (FDR<0.1) in this first of a kind study. Findings will need validation in enlarged cohorts.